# Star formation shut down by multiphase gas outflow in a galaxy at a redshift of 2.45

Sirio Belli[1✉], Minjung Park[2], Rebecca L. Davies[3,4], J. Trevor Mendel[4,5], Benjamin D. Johnson[2], Charlie Conroy[2], Chloë Benton[6], Letizia Bugiani[1], Razieh Emami[2], Joel Leja[7,8,9], Yijia Li[7,8], Gabriel Maheson[10,11], Elijah P. Mathews[7,8,9], Rohan P. Naidu[12], Erica J. Nelson[6], Sandro Tacchella[10,11], Bryan A. Terrazas[13] & Rainer Weinberger[14]

Large-scale outflows driven by supermassive black holes are thought to have a fundamental role in suppressing star formation in massive galaxies. However, direct observational evidence for this hypothesis is still lacking, particularly in the young universe where star-formation quenching is remarkably rapid[1–3], thus requiring effective removal of gas[4] as opposed to slow gas heating[5,6]. Although outflows of ionized gas are frequently detected in massive distant galaxies[7], the amount of ejected mass is too small to be able to suppress star formation[8,9]. Gas ejection is expected to be more efficient in the neutral and molecular phases[10], but at high redshift these have only been observed in starbursts and quasars[11,12]. Here we report JWST spectroscopy of a massive galaxy experiencing rapid quenching at a redshift of 2.445. We detect a weak outflow of ionized gas and a powerful outflow of neutral gas, with a mass outflow rate that is sufficient to quench the star formation. Neither X-ray nor radio activity is detected; however, the presence of a supermassive black hole is suggested by the properties of the ionized gas emission lines. We thus conclude that supermassive black holes are able to rapidly suppress star formation in massive galaxies by efficiently ejecting neutral gas.

We observed the galaxy COSMOS-11142 as part of the Blue Jay survey, a Cycle-1 James Webb Space Telescope (JWST) programme that targeted about 150 galaxies uniformly distributed in redshift $z$ ($1.7 < z < 3.5$) and stellar mass $M_*$ ($\log M_*/M_\odot > 9$). COSMOS-11142 is among the most massive targets, with $\log M_*/M_\odot = 10.9$, and is one of the 17 objects that are classified as quiescent according to the rest-frame UVJ colours[13]. This is confirmed by the JWST Near Infrared Spectrograph (NIRSpec) spectrum, shown in Fig. 1, which covers the rest-frame range from 3,000 Å to 1.4 μm and features several stellar absorption lines and relatively weak emission lines from ionized gas.

We fit stellar population models to the observed spectroscopy together with the photometry, which covers a wider wavelength range from the ultraviolet to the mid-infrared. The best-fit spectral model is shown in red in Fig. 1, and includes stellar light, absorption by dust and re-emission by dust at longer wavelengths, but does not include the contribution of gas. By analysing the difference between the data and the model, we detect several emission lines due to warm ionized gas: [O II], [Ne III], Hβ, [O III], Hα, [N II], [S II], [S III] and He I. We also detect three absorption lines, Ca II K, Ca II H and Na I D, that are not correctly reproduced by the stellar model. Unlike the other absorption lines visible in the spectrum, these three are resonant lines, meaning

that they can be produced both by stars and by intervening gas, because they involve transitions out of the atomic ground level. The detection of extra absorption in these lines is thus revealing the presence of cold gas. As the energy required to ionize Na I and Ca II (5.1 eV and 11.9 eV, respectively) is smaller than that required to ionize hydrogen (13.6 eV), these lines probe gas in the neutral atomic phase (that is, where hydrogen atoms are neutral).

We fit a Gaussian profile to each gas emission and absorption line, obtaining a measure of their flux, line width $\sigma$ and velocity offset $\Delta v$ with respect to the systemic velocity of the galaxy. We show a selection of lines in the left column of Fig. 2: a remarkable diversity of kinematics is apparent, with a wide range of measured line widths and velocity offsets. The presence of blueshifted lines, both in absorption and in emission, with velocity offsets of hundreds of kilometres per second is the unmistakable sign of a gas outflow.

We use a simple model of a biconical outflow, shown in the right column of Fig. 2, to qualitatively explain the kinematics of all the observed lines. The five rows of the figure illustrate five different types of line, classified according to their kinematics, which we discuss here from top to bottom. (1) Low-ionization lines such as [O II] have kinematics in agreement with those of the stellar population (that is, $\Delta v \approx 0$ km s$^{-1}$

[1]Dipartimento di Fisica e Astronomia, Università di Bologna, Bologna, Italy. [2]Center for Astrophysics | Harvard & Smithsonian, Cambridge, MA, USA. [3]Centre for Astrophysics and Supercomputing, Swinburne University of Technology, Hawthorn, Victoria, Australia. [4]ARC Centre of Excellence for All Sky Astrophysics in 3 Dimensions (ASTRO 3D), https://astro3d.org.au. [5]Research School of Astronomy and Astrophysics, Australian National University, Canberra, Australian Capital Territory, Australia. [6]Department for Astrophysical and Planetary Science, University of Colorado, Boulder, CO, USA. [7]Department of Astronomy and Astrophysics, The Pennsylvania State University, University Park, PA, USA. [8]Institute for Gravitation and the Cosmos, The Pennsylvania State University, University Park, PA, USA. [9]Institute for Computational and Data Sciences, The Pennsylvania State University, University Park, PA, USA. [10]Kavli Institute for Cosmology, University of Cambridge, Cambridge, UK. [11]Cavendish Laboratory, University of Cambridge, Cambridge, UK. [12]MIT Kavli Institute for Astrophysics and Space Research, Cambridge, MA, USA. [13]Columbia Astrophysics Laboratory, Columbia University, New York, NY, USA. [14]Leibniz Institute for Astrophysics, Potsdam, Germany. ✉e-mail: sirio.belli@unibo.it

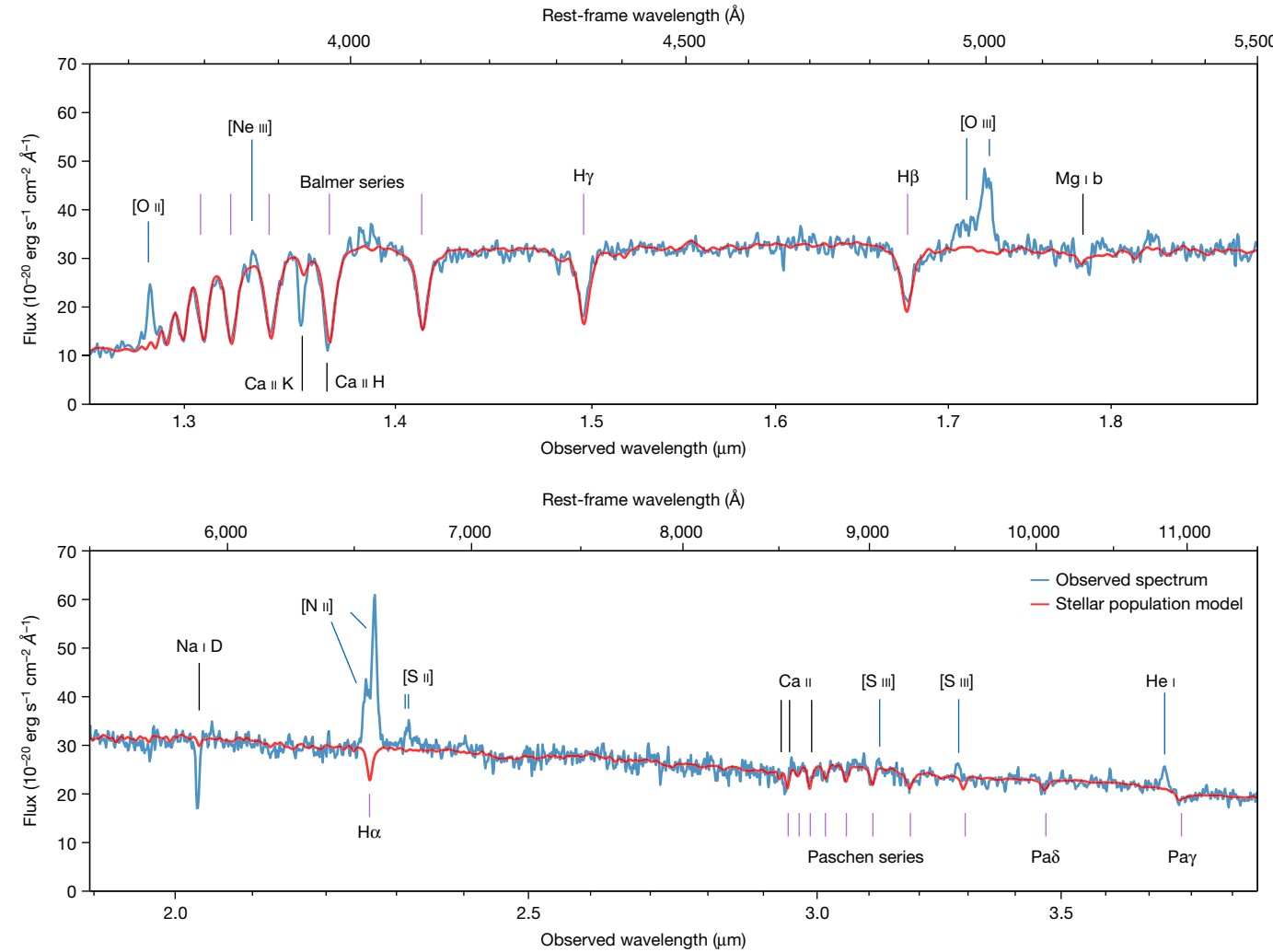

**Fig. 1 | JWST/NIRSpec spectrum of COSMOS-11142.** The best-fit model, which includes the contribution of stars and dust, is shown in red. Data and model are inverse-variance smoothed using a two-pixel window. Important absorption lines due to hydrogen (violet) and metals (black) and emission lines due to ionized gas (blue) are marked. The discrepancy between the stellar model and the data reveals the presence of substantial neutral gas (absorption by Ca II and Na I) and ionized gas (emission by O II, O III, N II and other species).

and $\sigma \approx 300$ km s$^{-1}$), suggesting that they originate in the galaxy and not in the outflow. (2) The neutral absorption lines Na I D and Ca II K are significantly blueshifted with $\Delta v \approx -200$ km s$^{-1}$, and are therefore tracing the foreground gas that is in the approaching side of the outflow. As this gas must remain neutral, it is probably farther out compared with the ionized gas, and we assume it has the shape of a thin shell. (3) Emission lines with a relatively high ionization energy, such as [O III], are also blueshifted ($\Delta v \approx -200$ km s$^{-1}$ to $\Delta v \approx -400$ km s$^{-1}$), and are thus likely to originate in the approaching side of the outflow. (4) A special case is [S III], which is also a high-ionization emission line but is observed to be at roughly systemic velocity ($\Delta v \approx 0$ km s$^{-1}$) and with a line width that is too broad to be produced by the gas in the galaxy ($\sigma \approx 600$ km s$^{-1}$); this emission is probably tracing both the approaching and the receding side of the outflow. The difference with the other high-ionization emission lines is due to the redder rest-frame wavelength of [S III], which makes it less prone to dust attenuation and allows us to see the full velocity distribution. The [O III] emission is thus blueshifted not because the outflow is asymmetric, but because its far, redshifted side is hidden by dust attenuation. (5) Finally, He I is the only emission line that is redshifted ($\Delta v \approx +400$ km s$^{-1}$) compared with the systemic velocity of the galaxy, implying that we are seeing the receding side of the outflow but not the approaching side. This peculiar behaviour (often seen for Lyman α) is due to resonant scattering, because the He I transition

involves a meta-stable state and can therefore be self-absorbed. As the meta-stable level can only be populated via recombination, the He I line traces the ionized gas. The overall picture emerging from the observations is that of an outflow that is present both on the foreground and on the background of the galaxy: this could be either a biconical or a spherical outflow.

From the observed line profiles, we derive high outflow velocities for the ionized gas, reaching up to about 1,700 km s$^{-1}$ in the case of [O III], which strongly suggests that the outflow is driven by an active galactic nucleus (AGN). The presence of an AGN is confirmed by the high [N II]/Hα and [O III]/Hβ line ratios[14].

Following standard modelling for the ionized[15,16] and neutral[17] phases, we measure $M_{out}$, the mass of gas in the outflow. The derived outflow masses are particularly sensitive to the assumptions made in the derivation, such as the electron number density (for the ionized phase) and the dust depletion together with the opening angle (for the neutral phase; see Methods for details). The uncertainties are therefore dominated by systematics, which we estimate to be about 0.7 dex for both the ionized and the neutral outflow mass. Dust depletion is particularly uncertain for calcium, and so we can use the Ca II K line only to derive a lower limit on the outflow mass. The resulting outflow velocities and masses are shown in Fig. 3a. The ionized outflow masses derived from four different emission lines are in remarkable agreement, which validates some

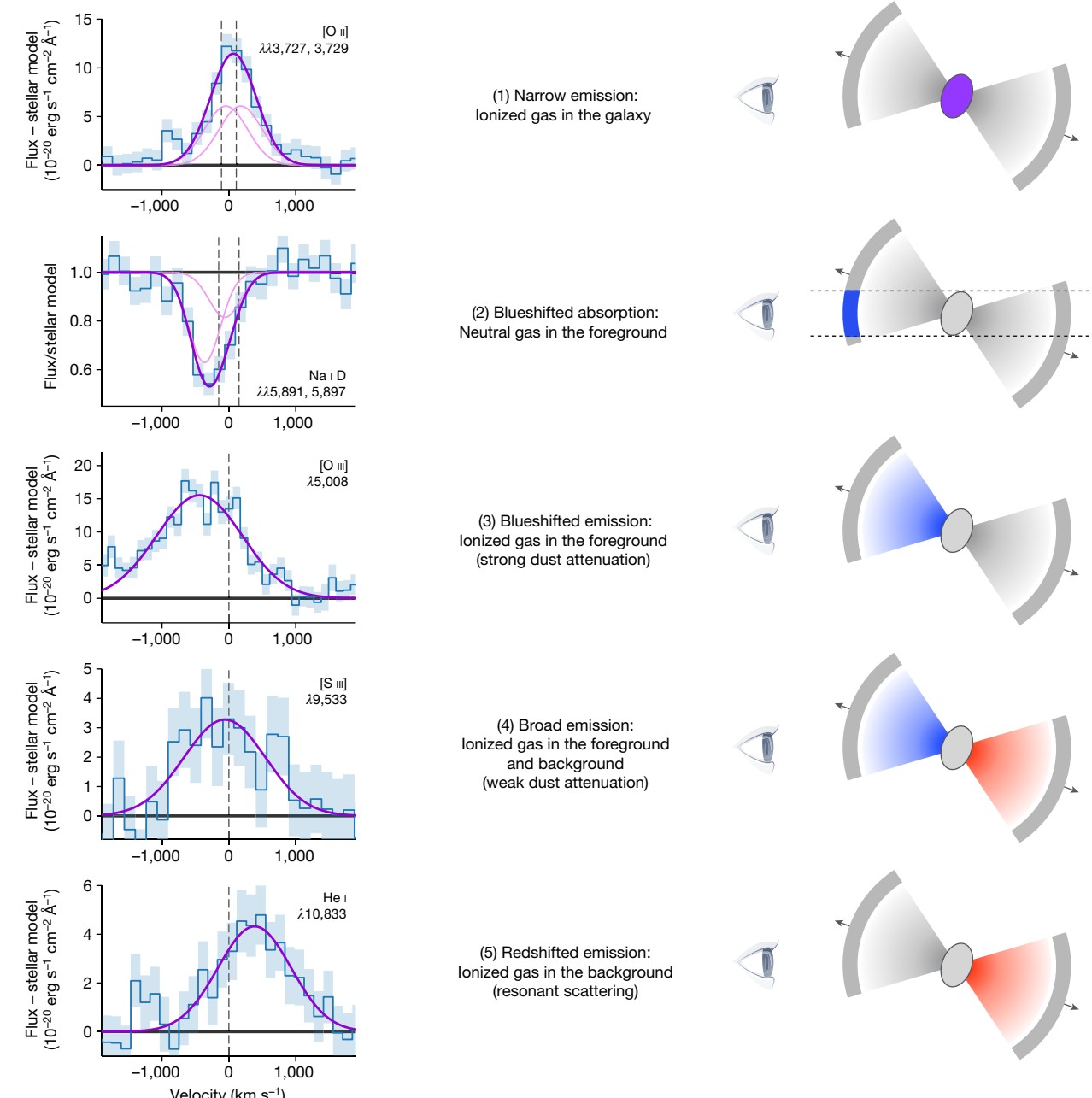

**Fig. 2 | Kinematics of gas emission and absorption lines.** Each row illustrates a different kinematic component of the gas, with the observations on the left and a cartoon on the right showing which spatial regions (highlighted in colour) are probed by the observations. In the left panels, Gaussian fits are shown in purple, with fits to the individual lines in doublets ([O II] and Na I D) shown in a lighter colour. For the emission lines, the difference between the observed flux and the best-fit stellar model is shown; for the absorption line, the ratio of the observed flux and the best-fit stellar model is shown. The zero of the velocity scale corresponds to the redshift of the stellar component measured from spectral fitting, and the dashed vertical lines mark the expected rest-frame location of the emission or absorption lines.

of the assumptions made in the modelling. Moreover, we find that the neutral outflow is slower but has a substantially larger mass, in qualitative agreement with observations of local galaxies[18].

Knowing the outflow mass and velocity allows us to calculate the mass outflow rate, $\dot{M}_{out} = M_{out} v_{out}/R_{out}$, where $R_{out}$ is the size of the outflow. We estimate $R_{out} \approx 3$ kpc from the NIRSpec data, in which we are able to spatially resolve the blueshifted [O III] emission. We then find a mass outflow rate $\dot{M}_{out} \approx 35\,M_\odot\,\mathrm{yr}^{-1}$ for the neutral outflow and $\dot{M}_{out} \approx 1\,M_\odot\,\mathrm{yr}^{-1}$ for the ionized outflow. The ratio between the two phases is large, but within the range measured in local outflows[18–22]. However, the mass outflow rate of COSMOS-11142 is an order of

magnitude larger than the typical values measured in local star-forming galaxies[23,24]. At high redshift, measurements of neutral outflows from Na I D absorption have been obtained for only a few quasars[25,26], but observations of ultraviolet or submillimetre lines tracing neutral gas reveal high mass outflow rates in star-forming galaxies and AGN systems[11].

To understand the role of the outflow in the evolution of COSMOS-11142, in Fig. 3b we show the star formation history of the galaxy, derived from our spectro-photometric fit. We find that the system is in a 'post-starburst' phase: it formed most of its stellar mass in a rapid and powerful starburst about 300 Myr before the observations, and

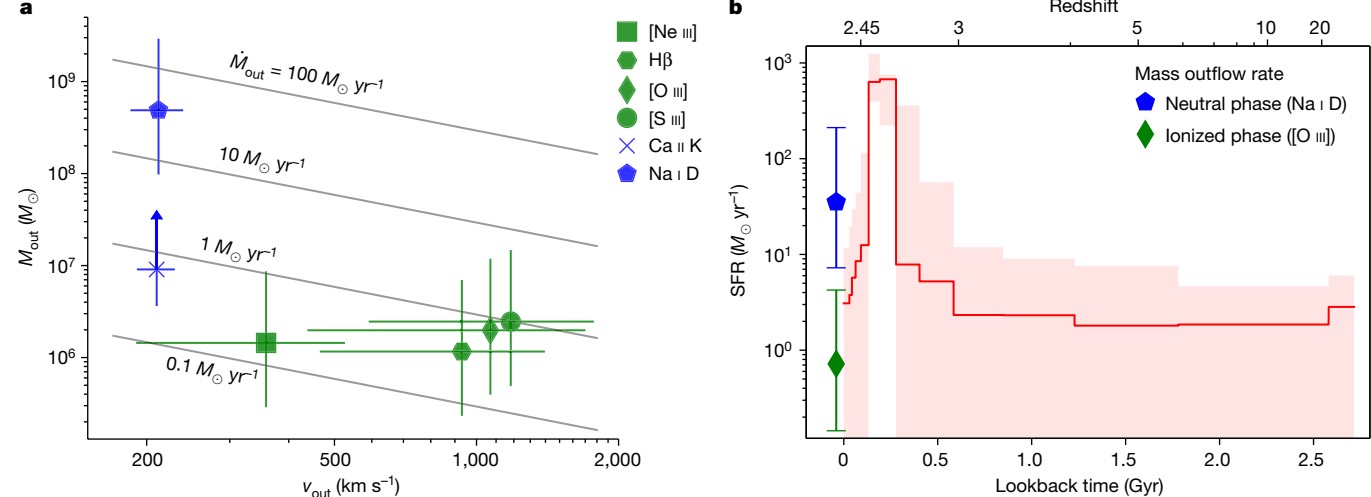

**Fig. 3 | Physical properties of the outflow and comparison with the star formation history. a**, Outflow mass versus velocity derived for individual lines, probing neutral and ionized gas. The vertical error bars show the 0.7-dex systematic uncertainty on the outflow mass measurements and the horizontal error bars reflect the systematic uncertainty on the definition of outflow velocity. The Ca II K outflow mass is derived assuming no depletion onto dust, and is therefore a lower limit. The diagonal lines are at constant mass outflow rate assuming $\dot{M}_{out} = M_{out}\, v_{out}/R_{out}$. **b**, Star formation history (red line) and 95% credible region (shaded area) derived from fitting the spectroscopic and photometric data. The mass outflow rate is shown for neutral and ionized gas at a lookback time of zero (that is, the epoch at which the galaxy is observed). The mass outflow rate for the neutral gas is substantially higher than the residual star formation rate, implying that the outflow is able to strongly suppress the star formation activity.

then experienced a rapid quenching of the star formation rate by two orders of magnitude. These remarkably rapid formation and quenching timescales are not seen in the local universe, but are common among massive systems at $z \approx 1$–2 (refs. 1–3), and represent the only way to form quiescent galaxies at even higher redshift[27] owing to the younger age of the universe. According to the star formation history, the rate at which COSMOS-11142 is currently forming stars is between $1\,M_\odot\,\mathrm{yr}^{-1}$ and $10\,M_\odot\,\mathrm{yr}^{-1}$; we obtain consistent estimates from ionized emission lines and infrared emission. The system is therefore in the middle of quenching, about 1 dex below the main sequence of star formation[28], but still above the bulk of the quiescent population[29,30]. By comparing the mass outflow rate with the current star formation rate, we conclude that the ionized outflow is weak, whereas the neutral outflow is very strong. This comparison shows that the ionized outflow is irrelevant in terms of gas budget, whereas the neutral outflow is able to substantially affect the star formation rate by ejecting cold gas before it can be transformed into stars. We thus conclude that the observed outflow probably has a key role in the rapid quenching of COSMOS-11142. Given the low outflow velocity, $v_{out} \approx 200\,\mathrm{km\,s^{-1}}$, it is possible that most of the neutral gas is not able to escape the galaxy. In this case, heating of the halo gas by radio-mode AGN feedback is probably required to maintain this galaxy quiescent over a Hubble timescale. However, this does not change our main conclusion, as radio-mode AGN feedback alone is unable to explain the observed rapidity of quenching.

Despite the highly effective feedback in action, COSMOS-11142 is not detected in publicly available X-ray or radio observations. This suggests that AGN samples selected at those wavelengths do not necessarily probe the galaxy population in which feedback is being most effective. Such samples are usually biased towards powerful AGNs, which tend to live in gas-rich, star-forming galaxies[31,32]. Strong neutral outflows similar to the one detected in COSMOS-11142 may in fact be present in the majority of massive galaxies[33], which often have emission line ratios consistent with AGN activity despite the lack of X-ray emission[7,34]. Among massive galaxies, AGN-driven outflows may be particularly important for the post-starburst population, which is very likely to host emission lines with high [N II]/Hα ratio[3,30]. Moreover, the detection of blueshifted gas absorption in post-starburst galaxies at $z \approx 1$

(refs. 35,36) suggests that neutral outflows are frequent during this specific evolutionary phase. The rapid quenching of massive galaxies at $z > 1$ may thus be fully explained by the AGN-driven ejection of cold gas.

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

## Methods

### JWST spectroscopy

The Blue Jay survey is a Cycle-1 JWST programme (GO 1810) that observed about 150 galaxies at $1.7 < z < 3.5$ in the COSMOS field with the NIRSpec Micro-Shutter Assembly (MSA). Each galaxy was observed with the three medium-resolution gratings, covering the 1–5 μm wavelength range at a spectral resolution of $R \approx 1,000$. The target sample was drawn from a catalogue[37] based on Hubble Space Telescope (HST) data. The selection was designed in a way to obtain a roughly uniform coverage in redshift and mass, and the sample is unbiased above a redshift-dependent stellar mass of $10^{8.7}$–$10^{9.3} M_\odot$. COSMOS-11142 (the ID is from ref. 37) was observed in December 2022 for a total of 13 hours in G140M, 3.2 hours in G235M and 1.6 hours in G395M. A slitlet made of four MSA shutters (shown in Extended Data Fig. 1) was placed on the target and the observations employed a two-point A–B nodding pattern along the slit. We reduced the NIRSpec data using a modified version of the JWST Science Calibration Pipeline v1.10.1, using version 1093 of the Calibration Reference Data System. Before combining the data, we visually inspected and excluded any remaining artefacts in the individual exposures. The galaxy one-dimensional spectrum was then optimally extracted. For more details on the Blue Jay survey and the data reduction, see S.B. et al. (manuscript in preparation).

### JWST imaging

JWST imaging of COSMOS-11142 is available from the PRIMER survey (GO 1837, principal investigator J. Dunlop) in several bands: F090W, F115W, F150W, F200W, F277W, F356W, F410M and F444W with the Near-Infrared Camera (NIRCam); and F770W and F1800W with the Mid-Infrared Instrument (MIRI). We performed aperture photometry on the MIRI data and applied a point-source correction derived from WebbPSF[38]. For the NIRCam data, which have a higher resolution and sensitivity, we fit the surface brightness profile of COSMOS-11142, independently in each band, using Forcepho (B.D.J. et al., manuscript in preparation). We model the galaxy using a single Sersic profile, convolved with the point spread function, and explore the posterior distribution via Markov chain Monte Carlo. This yields a multi-band set of photometric and structural measurements. Extended Data Fig. 1 shows the data and the model for F200W, which is the short-channel band with the highest signal-to-noise ratio (212), where we measure an effective (half-light) radius $R_e = 0.075''$, corresponding to 0.6 kpc, a Sersic index $n = 2.6$ and an axis ratio $q = 0.5$. Although the formal errors are small, and the measurements are probably dominated by systematic uncertainties, we can robustly conclude that the galaxy is compact (yet well resolved in the NIRCam data) and elongated. These results are qualitatively unchanged when considering the other NIRCam bands. The residuals are small, implying that a single Sersic profile is a good description of the galaxy morphology and ruling out the presence of a major merger, a close companion or bright point-source emission from a type-1 AGN.

### Spectral fitting

We characterize the stellar population and dust properties of COSMOS-11142 by fitting models to the observed spectroscopic and photometric data. We use the Bayesian code Prospector[39] and follow the approach explained in refs. 3,40. We adopt the synthetic stellar population library FSPS[41,42], the Mesa Isochrones and Stellar Tracks (MIST) isochrones[43], the C3K spectral library[44] and the Chabrier initial mass function[45]. The galaxy stellar population is described by stellar mass, redshift, velocity dispersion, metallicity and a non-parametric star formation history with 14 bins. The bins are logarithmically spaced except for the youngest one (0–30 Myr), and the oldest, which is a narrow bin placed at the age of the Universe, providing the possibility of a maximally old population. We adopt a continuity prior that disfavours abrupt changes in the star formation history (see ref. 46 for

details). The model also includes attenuation by dust, described by three parameters (attenuation in the $V$ band $A_V$, dust index and extra attenuation towards young stars[47,48]), and dust emission, implemented with three free parameters describing the infrared emission spectrum[49]. We assume that the total amount of energy absorbed by dust is then re-emitted in the infrared. We do not include the contribution from gas or AGNs.

To fit a single model to both the JWST spectroscopy and the multi-wavelength photometry, it is necessary to include important systematic effects, as described in ref. 39. We add one parameter describing the fraction of spectral pixels that are outliers, and one 'jitter' parameter that can rescale the spectroscopic uncertainties when necessary to obtain a good fit. The best-fit value for the jitter parameter is 2.05, suggesting that the NIRSpec data reduction pipeline underestimates the spectral uncertainties. In our subsequent analysis of the emission and absorption lines, we apply this jitter term to the error spectrum, to obtain a more accurate estimate of the uncertainties on our results.

We also adopt a polynomial distortion of the spectrum to match the spectral shape of the template, to allow for imperfect flux calibration and slit-loss corrections (particularly important in this case as the shutter covers only a fraction of the galaxy). In practice, this is equivalent to normalizing the continuum and considering only the small-scale spectral features such as breaks and absorption lines. In turn, this procedure yields an accurate flux calibration for the JWST spectrum, if we assume that the emission probed by the MSA shutter is just a rescaling of the emission probed by the photometry (that is, we are neglecting strong colour gradients). This yields a slit-loss correction of about two, with a small dependence on wavelength. The spectrum shown in Fig. 1 has been calibrated in this way, and we adopt this calibration also in subsequent analysis of absorption and emission lines.

The model has a total of 25 free parameters, and to fully explore the posterior distribution we use the nested sampling package dynesty[50]. We fit the model to the observed NIRSpec spectroscopy and to the broadband data (shown in Extended Data Fig. 2). In the spectrum, we mask ionized gas emission lines, including Hα and Hβ, and the resonant absorption lines Na I D, Ca II H and Ca II K. For the photometry, we make use of our measurements from MIRI and NIRCam data (excluding F356W and F410M because the data reduction pipeline flagged most of the galaxy pixels as outliers). We also adopt archival photometry from HST, measured from data taken with the Advanced Camera for Surveys (ACS) and the Wide-Field Camera 3 (WFC3) instruments[37]. These measurements are clearly offset from the JWST NIRCam photometry, probably because they have been measured using a different method (photometry within a fixed aperture). From comparing the HST F160W and the JWST F150W bands, which cover a very similar wavelength range, we determine that the HST fluxes are overestimated by 26%. We correct all the HST points for this offset, and add in quadrature a 26% relative error to the HST uncertainties.

The best-fit model is shown in red in Fig. 1 and Extended Data Fig. 2: the same model is able to simultaneously reproduce the spectroscopy and the photometry. The fit yields a stellar mass $\log M_*/M_\odot = 10.9$, a stellar velocity dispersion $\sigma = 273 \pm 13$ km s$^{-1}$ and a metallicity $[\text{Fe/H}] = 0.16 \pm 0.05$; the resulting star formation history is shown in Fig. 3. The galaxy is relatively dusty, with $A_V = 1.5 \pm 0.1$, which is higher than what found in quiescent systems and may be related to the rapid quenching phase in which it is observed. We find that the main results of our analysis do not change when excluding the HST photometry, using a smaller wavelength range for the spectroscopic data or changing the order of the polynomial distortion.

### Absorption and emission lines

To analyse the gas emission and absorption lines, we first mask these features when running Prospector, and then fit the residuals using Gaussian profiles convolved with the instrumental resolution. The results of the Gaussian fits are listed in Extended Data Table 1.

We show the resonant absorption lines Ca II H, Ca II K and Na I D in Extended Data Fig. 3. These lines are also present in the best-fit stellar model, but the observed absorption is both much stronger and clearly blueshifted, making it easier to study the neutral gas. We also note that the Ca II H line lies on top of the Balmer Hε absorption line, which however is not resonant and is therefore present only in the stellar spectrum.

As the effect of neutral gas absorption is multiplicative, when fitting Gaussian profiles we consider the ratio of the observed spectrum to the stellar model. We model Ca II H and Ca II K with a Gaussian profile each, assuming the same width and velocity offset. Owing to the faintness of Ca II, it is difficult to measure the doublet ratio, so we fix it to 2:1, appropriate for optically thin gas[51], which seems to reproduce well the data. We independently fit the Na I D doublet, which is unresolved, using two Gaussians with the same width and velocity offset, and fixing their equivalent width ratio to the optically thin value of 2:1. We verified that the results do not change in a substantial way when leaving the doublet ratio free.

The observed line profiles do not warrant the modelling of multiple kinematic components; however, we consider the possibility that the observed absorption consists of the sum of a blueshifted and a systemic component. For example, this could be the case if our model underestimates the stellar absorption. We use the Alf stellar population synthesis code[52,53] to assess how the observed absorption lines vary when changing the abundance of individual elements in the stellar populations. We conclude that this effect is negligible: for a Na abundance that is twice the solar value, the extra absorption in Na I D would be only 8% of the equivalent width we observe in COSMOS-11142. A systemic component could also arise from neutral gas that is in the galaxy and not in the outflow. We find this unlikely because of the increased importance of the molecular phase, at the expense of the neutral phase, for the gas reservoir of galaxies at high redshift[54]; and also based on a study of Na I D absorption in the Blue Jay survey, where neutral gas is mostly associated with outflows and not with the gas reservoir, even in star-forming galaxies[33]. Nonetheless, we repeat the fit for Ca II K, which is well detected and not blended, adding a Gaussian component fixed at systemic velocity. In this case, we find that the equivalent width of the blueshifted component would be reduced by $(41 \pm 9)\%$. Finally, an additional source of systematic uncertainty could be the presence of resonant redshifted emission from Na I D, which would fill up the red side of the Na I D absorption line and thus yield a slight underestimate of the equivalent width and a slight overestimate of the outflow velocity. Resonant Na I D emission is rarely seen in the local universe[19], but the presence of resonant He I emission suggests this could be a possibility in COSMOS-11142.

Similarly to what was done for the absorption lines, we fit Gaussian profiles to all the detected emission lines, which are shown in Fig. 2 and Extended Data Fig. 4. Given the additive nature of the emission, here we consider the difference between the observed spectrum and the stellar model. The emission line kinematics present a wide diversity and thus require a sufficiently flexible model. For this reason, we try to fit each line independently, imposing physically motivated constraints only when necessary because of blending and/or low signal-to-noise ratio. The most complex case is the blend of Hα with the [N II] doublet. We model all three lines simultaneously, assuming that they have identical velocity offset and dispersion, and we also fix the [N II] doublet ratio to the theoretical value of 3:1. The resulting best fit, shown in Extended Data Fig. 4a, approximately reproduces the data, even though some discrepancies in the flux peaks and troughs are visible. If we add a broad Hα component to the model, we obtain a marginally better fit (Extended Data Fig. 4b). The broad component has a velocity dispersion $\sigma \approx 1{,}200$ km s$^{-1}$, and may be due to a faint broad line region[27]. However, we obtain a nearly identical fit if we adopt a broad component for the [N II] lines as well, in which case the broad component has $\sigma \approx 800$ km s$^{-1}$ and would be due to the outflow. We conclude that the lines are too blended to robustly constrain their kinematics (although the line fluxes remain consistent across the different fits, within the uncertainties), and we adopt the simplest model, consisting of three single Gaussian profiles. The velocity dispersion obtained with this model is large (465 km s$^{-1}$), suggesting the presence of an outflow component in these lines. Other blended doublets in our spectrum are [O II] and [S II]; for each of them, we fix the doublet flux ratio to 1:1, which is in the middle of the allowed range, and assume that the two lines in the doublet have identical kinematics. Finally, given the low signal-to-noise ratio of the Hβ and [S II] lines, we also decide to fix their kinematics to those derived for Hα and [N II], as they all have similar ionization energy. We note that if we instead fit the Hβ line independently, we measure a blueshift, which would be consistent with the emission originating in the approaching side of the outflow, but the line is too faint for drawing robust conclusions. To summarize, we assume the same kinematics for the low-ionization lines Hα, Hβ, [N II] and [S II]; obtaining a dispersion consistent with the presence of an outflow. The other low-ionization line [O II] represents an exception, as it has a much smaller line width that does not show evidence of outflowing material; however, we analyse the [O II] morphology in the two-dimensional spectrum and find the clear presence of a faint outflow component (as discussed below). For the high-ionization lines [O III], [Ne III] and [S III], we leave the kinematics free when fitting. We measure a blueshift for [O III] and [Ne III], although the latter has a much lower velocity shift. This discrepancy may simply be due to the low signal-to-noise ratio of [Ne III]; we also try to fix its kinematics to that of [O III] but the fit is substantially worse. Interestingly, for [S III], we do not measure a statistically significant blueshift, but the velocity dispersion is very large, probably tracing the contribution of both the receding and the approaching sides of the outflow. Finally, He I shows a clear redshift, which is the sign of resonant scattering. We note that [S III] and He I have similar wavelength and require similar ionization energy, meaning that they probe the same type of gas conditions. One important difference is that the blue side of He I experiences resonant scattering; however, the red side is not affected by this phenomenon. The red side of [S III] and He I must therefore trace almost exactly the same type of gas, which is found in the receding outflow. Our interpretation of the observed kinematics for these two lines is thus self-consistent.

Clearly, the emission lines in COSMOS-11142 present a remarkable complexity. Our goal is to estimate the properties of the ionized outflow to understand its role in the evolution of the galaxy; a detailed study of the ionized emission lines is beyond the scope of this work. For this reason, we avoid a decomposition into broad and narrow component for the brightest emission lines, and choose to fit each line independently, when possible. We have also tested a global fit, where all lines except for He I and [O III] have the same kinematics; and also a model with separate kinematics for high- and low-ionization lines. The result of these tests is that the fluxes for the lines used in the calculation of the ionized outflow mass ([O III], [Ne III], [S III] and Hβ) are remarkably stable, with different assumptions causing differences in the fluxes that are always smaller than the statistical uncertainties.

## Star formation rate

The star formation rate of COSMOS-11142 is a key physical quantity, as it is used both to confirm the quiescent nature of the system and as a comparison with the mass outflow rate. We employ several methods to estimate the star formation rate, obtaining consistent results. The Prospector fit gives a star formation rate of about 3 $M_\odot$ yr$^{-1}$ in the youngest bin (0–30 Myr), with an uncertainty of a factor of 3; considering the average star formation over the past 100 Myr we obtain an upper limit of 10 $M_\odot$ yr$^{-1}$. An alternative, independent method relies on the hydrogen recombination emission lines; because of the contribution from the outflow to the observed flux, this method can only yield an upper limit on the star formation rate. We first correct the measured emission line fluxes for dust attenuation using the result of the Prospector fitting (including the extra attenuation towards young stars); we then use the standard conversion[55] applied to the measured Hα flux,

obtaining an upper limit of $10\,M_\odot\,\mathrm{yr}^{-1}$. A similar upper limit is obtained using the observed Hβ flux. It is possible that this method misses heavily dust-obscured regions hosting a starburst, as it is sometimes found in local post-starburst galaxies[56]. We check for this possibility by using redder hydrogen emission lines, which can be used to probe deeper into the dust: although we do not detect any of the Paschen lines in emission, we can place a 3σ upper limit to the Paschen γ flux of $2.2 \times 10^{-18}\,\mathrm{erg\,s^{-1}\,cm^{-2}}$, yielding an upper limit on the star formation of $28\,M_\odot\,\mathrm{yr}^{-1}$. The absence of substantial star formation in dust-obscured regions is also confirmed by the lack of strong mid-infrared emission. COSMOS-11142 is not detected in the Spitzer 24-μm observations from the S-COSMOS survey[57], yielding a 3σ upper limit of $38\,M_\odot\,\mathrm{yr}^{-1}$, according to the measurements and assumptions detailed in ref. 58. However, this estimate assumes that dust is heated exclusively by young stars—a bias known to lead to an overestimate of the star formation rate for quiescent galaxies, in which most of the heating is due to older stars[59]. The more sensitive JWST/MIRI observations detect the galaxy at 18 μm, and the measured flux is fully consistent with the best-fit Prospector model, as shown in Extended Data Fig. 2, thus confirming the lack of substantial star formation hidden by dust.

## Ionized outflow

We detect clear signs of ionized outflow as blueshifted emission in [O III] and [Ne III], and as a broad emission in [S III]. We can independently estimate the outflow properties using each of these emission lines. For a generic element X, the outflow mass can be written as a function of the observed line luminosity $L$ (refs. 15,16):

$$M_{\mathrm{out}} = \frac{1.4\,m_{\mathrm{p}}\,L}{n_{\mathrm{e}}\,10^{[\mathrm{X/H}]}\,(n_{\mathrm{X}}/n_{\mathrm{H}})_\odot\,j}, \qquad (1)$$

where $m_{\mathrm{p}}$ is the proton mass, $n_{\mathrm{e}}$ is the electron density, [X/H] is the logarithmic elemental abundance in the gas relative to the solar value $(n_{\mathrm{X}}/N_{\mathrm{H}})_\odot$ (which we take from ref. 60) and $j$ is the line emissivity. In this relation we neglect a factor $\langle n_{\mathrm{e}}^2\rangle/\langle n_{\mathrm{e}}\rangle^2$, and we assume that all the atoms of the element X are found in the ionization stage responsible for the observed line (this is consistent with the observation of strong [O III] emission in the outflow but nearly absent [O II]; the other outflow-tracing emission lines have comparable excitation energy). We calculate the emissivity for each line using pyNeb[61] with standard assumptions (density $500\,\mathrm{cm}^{-3}$ and temperature $10^4\,\mathrm{K}$). We further assume that the gas in the outflow has solar metallicity, given the high stellar mass of the galaxy and the result of the Prospector fit. The electron density cannot be reliably derived from the flux ratio of the poorly resolved [S II] and [O II] doublets. We make the standard assumption of $n_{\mathrm{e}} = 500\,\mathrm{cm}^{-3}$ (ref. 16); however, we note that this value is highly uncertain. As local studies using different methods find a wide range of values, $\log n_{\mathrm{e}} \approx 2$–3.5 (ref. 62), we assign a systematic uncertainty of 0.7 dex (a factor of 5 in each direction) to the assumed value of $n_{\mathrm{e}}$. For the blueshifted lines, we multiply the observed luminosity by two to account for the receding side hidden by dust. The resulting outflow masses are listed in Extended Data Table 2 (see also Fig. 3a). We also include an estimate of the outflow mass based on the Hβ line, assuming it is entirely originating in the outflowing gas. The four estimates of the outflow mass are in agreement with each other to better than a factor of two, which is remarkable given that the mass derived from Hβ does not depend on assumptions on the ionization stage and the gas metallicity. However, all four lines depend in the same way on $n_{\mathrm{e}}$. The uncertainty on the outflow mass measurement is therefore dominated by the assumed $n_{\mathrm{e}}$.

To calculate the mass outflow rate, we assume that the ionized gas is distributed, on each side of the outflow, in a mass-conserving cone that is expanding with velocity $v_{\mathrm{out}}$, independent of radius[17]. In this case, the mass outflow rate can be easily derived to be $\dot{M}_{\mathrm{out}} = M_{\mathrm{out}}\,v_{\mathrm{out}}/R_{\mathrm{out}}$. If the outflow is in a narrow cone directed towards the observer, then the velocity shift observed in blueshifted lines ([O III] and [Ne III]) corresponds to the outflow velocity: $v_{\mathrm{out}} = |\Delta v|$. However, if the emitting gas spans a range of inclinations, then the intrinsic outflow velocity corresponds to the maximum observed value, which is often defined as $v_{\mathrm{out}} = |\Delta v| + 2\sigma$ (refs. 10,17). As we do not know the opening angle and inclination of the outflow, we adopt an intermediate definition: $v_{\mathrm{out}} = |\Delta v| + \sigma$, and take σ as a measure of the systematic uncertainty, so that both scenarios are included within the error bars. For emission lines that are not blueshifted ([S III] and Hβ), instead, we simply take $v_{\mathrm{out}} = 2\sigma$ and use σ as a measure of the systematic uncertainty.

To constrain the outflow size $R_{\mathrm{out}}$, we analyse the two-dimensional NIRSpec spectrum around the location of the [O III] λ5,008 line. We first construct the spatial profile by taking the median flux along each spatial pixel row; then we subtract this profile, representing the stellar continuum, from the data, revealing a clearly resolved [O III] line (Extended Data Fig. 5). The emission line morphology is complex, consisting of a fast component in the galaxy centre and two slower components extending several pixels along the spatial direction on either side of the galaxy; all components are blueshifted, and are therefore not associated with a large-scale rotating disk. We measure the extent of the larger components to be approximately 4 spatial pixels, that is, 0.4″; we detect weak extended blueshifted emission also in the [O II], [N II] and Hα emission lines, confirming the presence of ionized gas about 0.4″ from the centre. We interpret this as the maximum extent of the ionized outflow, and we thus adopt $R_{\mathrm{out}} \approx 0.4″ \approx 3$ kpc. The line morphology in the two-dimensional spectrum is consistent with our physical model of the outflow: most of the emission comes from the central regions because they are denser (in a mass-conserving cone the gas density scales inversely with the square of the radius[17]), and the line blueshift is progressively weaker in the outer regions owing to projection effects[16]. However, it is also possible that we are seeing one fast, small-scale outflow together with a separate slow, large-scale outflow. If the size of the fast outflow is smaller than our estimate, the ionized outflow rate would then be correspondingly larger. Also, if the outflow is strongly asymmetric, then the slit-loss correction derived for the stellar emission may not be appropriate, which would introduce a bias in our line flux measurements and gas masses.

With the adopted value for the outflow size, we are able to derive ionized outflow masses, which are in the range $0.2$–$1\,M_\odot\,\mathrm{yr}^{-1}$. The outflow properties estimated from the four different lines are shown in Fig. 3a: it is clear that the different tracers give consistent results, particularly when considering the large systematic uncertainties in the outflow mass. Also, the order of magnitude of the mass outflow rate appears to be nearly independent of the exact definition of the outflow velocity: for any reasonable value of the velocity, the mass outflow rate of the ionized gas is unlikely to go substantially beyond about $1\,M_\odot\,\mathrm{yr}^{-1}$.

Despite the large size of the outflow, most of the emission comes from the central, denser regions, and can therefore be heavily attenuated by the dust present in the galaxy. We can estimate the effect of dust attenuation on different lines, using the modified Calzetti extinction curve with the best-fit parameters from spectral fitting. It is difficult to estimate the extra attenuation towards nebular emission because of the complex morphology of the outflow; we consider a wide range from 1 (that is, stars and gas are attenuated equally) to 2, which is the canonical value for starburst galaxies. At the [O III] wavelength, the emission from gas behind the galaxy is attenuated by a factor of about 5–27, explaining why we do not observe a redshifted component. The attenuation decreases to a factor of about 3–9 at the Hα wavelength, and is only a factor of about 2–3.5 at the [S III] wavelength (which indeed has an asymmetric profile, visible in Fig. 2, with some flux missing in the redshifted part probably owing to differential dust extinction).

## Neutral outflow

We use the observed Na I D and Ca II K lines to constrain the properties of the neutral outflow. The first step is to derive the Na I and

Ca II column density from the observed equivalent widths (listed in Extended Data Table 1), which can be done easily in the optically thin case[51], yielding $N_{NaI} = 2.2 \times 10^{13}$ cm$^{-2}$ and $N_{CaII} = 3.6 \times 10^{13}$ cm$^{-2}$. These should be considered lower limits, as even small deviations from the optically thin case can substantially increase the column density corresponding to the observed equivalent width. If the outflow is clumpy, and its covering fraction is less than unity, then the true equivalent width would be larger than the observed one. The observed depths of the two Ca II lines can be used to constrain the covering fraction[63]: in our case, the data are consistent with a covering fraction of unity, but with a large uncertainty due to the low signal-to-noise of Ca II H. A hard lower limit can also be obtained from the maximum depth of any of the neutral gas absorption lines; thus, the covering fraction must be larger than 50%. For simplicity, here we assume a covering fraction of unity; if the true value were smaller by a factor of two, then the column density would be larger by a factor of two, and the mass outflow rate, which depends on their product, would remain the same.

The next step consists in inferring the hydrogen column density. For Na I, we can write[19,64]:

$$N_H = \frac{N_{NaI}}{(1-y)\, 10^{[Na/H]}\, (n_{Na}/n_H)_\odot\, 10^b}, \tag{2}$$

where $y$ is the sodium ionization fraction, and $10^b$ represents the depletion of sodium onto dust. For consistency with local studies, we make the standard assumption $1-y = 0.1$ (ref. 64), meaning that only 10% of the sodium is in the neutral phase; it is likely that the true value is even lower[19], which would increase the derived column density and therefore the outflow mass. We also assume solar metallicity for the gas, and take the canonical values[65] for solar abundance, $\log(n_{Na}/n_H)_\odot = -5.69$, and dust depletion in the Milky Way, $b = -0.95$. We obtain a hydrogen column density $N_H = 9.6 \times 10^{20}$ cm$^{-2}$. The systematic uncertainty on this result is dominated by the observed scatter in the dust depletion value, which is 0.5 dex (ref. 66).

The calcium column density is more difficult to interpret because calcium, unlike sodium, shows a highly variable depletion onto dust as a function of the environment[67,68]. Observations of Milky Way clouds show very high dust depletion (up to 4 dex) for calcium at the high column density that we measure, which would imply a hydrogen column density that is about 20 times higher than what measured from Na I. This discrepancy is probably caused by the presence of shocks in the outflow, which can destroy dust grains and decrease the depletion of calcium. Thus, we can only derive a lower limit on the hydrogen column density by assuming that calcium is not depleted at all, and we obtain $N_H > 1.8 \times 10^{19}$ cm$^{-2}$. We have neglected the ionization correction as most of the calcium in the neutral gas is expected to be in the form of Ca II (ref. 69).

To derive the outflow mass, we assume that the neutral gas forms an expanding shell outside the ionized outflow. This is consistent with local observations[19] and with the idea that the neutral gas should be farther out from the ionizing source. In principle, a direct measurement of the neutral outflow size would require a background source extending far beyond the galaxy size; a source that does not exist in our case. However, this rare configuration has been observed for J1439B, a galaxy with similar mass, redshift and star formation rate to those of COSMOS-11142, which happens to be located near the line-of-sight to a background luminous quasar[70]. Neutral gas associated with J1439B and probably belonging to an AGN-driven outflow has been observed in absorption in the spectrum of the quasar, which is 38 projected kpc from the galaxy. This rare system suggests that AGN-driven neutral outflows from quenching galaxies can extend significantly beyond the stellar body of their host galaxies.

Given the large size of the outflow compared with the size of the stellar emission, we are probably detecting neutral gas that is moving along the line of sight, as shown in the cartoon in Fig. 2, and therefore we assume $v_{out} = |\Delta v|$. For a shell geometry, the outflow mass and mass rate are[17]:

$$M_{out} = 1.4\, m_p\, \Omega\, N_H\, R_{out}^2, \tag{3}$$

$$\dot{M}_{out} = 1.4\, m_p\, \Omega\, N_H\, R_{out}\, v_{out}, \tag{4}$$

where $\Omega$ is the solid angle subtended by the outflow. On the basis of the results of local studies[17] and on the incidence of neutral outflows in the Blue Jay sample[33], we assume an opening angle of 40% of the solid sphere, that is, $\Omega/4\pi = 0.4$. We consider the systematic uncertainty on $\Omega$ to be a factor of 2.5 in each direction, ranging from a narrow opening to a spherical and homogeneous outflow. Combined with the uncertainty on the calcium dust depletion, this gives a total uncertainty on the outflow mass of about 0.7 dex, similar to that on the ionized outflow mass (but due to a different set of assumptions).

The neutral mass outflow rate estimated from the Na I D line is 35 $M_\odot$ yr$^{-1}$, between 1 and 2 orders of magnitude larger than what is measured for the ionized phase. In addition to the 0.7-dex uncertainty owing to the opening angle and the dust depletion, there are two additional assumptions that could influence this result. First, we ignored a possible contribution to Na I D from a systemic component of neutral gas not associated with the outflow, which would decrease the measured column density by 41% (adopting the value derived for Ca II K). Second, we assumed that the radius of the neutral outflow coincides with that of the ionized outflow; in principle, the neutral outflow radius could be as small as the galaxy effective radius, which is 0.6 kpc (it cannot be smaller than this because the covering fraction must be larger than 50%, meaning that the neutral gas must be in front of at least 50% of the stars in the galaxy). If the neutral outflow were this small, the gas would be detected in the full range of inclinations, and the outflow velocity would be $v_{out} = |\Delta v| + 2\sigma$. If we make the most conservative choice on both the systemic component of Na I D and on the outflow radius, we obtain a mass outflow rate for the neutral phase $\dot{M}_{out} = 11\, M_\odot$ yr$^{-1}$. This is still one order of magnitude larger than the ionized mass outflow rate.

## Physical nature of the outflow and properties of the AGN

Owing to the low star formation rate of COSMOS-11142, it is unlikely that the outflow is driven by star formation activity. We can also rule out a star-formation-driven fossil outflow: the travel time to reach a distance of $R_{out} \approx 3$ kpc at the observed velocity is less than 10 Myr, which is much smaller than the time elapsed since the starburst phase, according to the inferred star formation history. Moreover, the ionized gas velocity we measure is substantially higher than what is observed in the most powerful star-formation-driven outflows at $z \approx 2$ (ref. 71). Another possibility could be that the outflowing material actually consists of tidally ejected gas due to a major merger[72,73]. However, the lack of tidal features or asymmetries in the near-infrared imaging, together with the high velocity of the ionized gas, rule out this scenario.

The only reasonable explanation is, therefore, that the observed outflow is driven by AGN feedback. This is confirmed by the emission line flux ratios, which we show in Extended Data Fig. 6. COSMOS-11142 occupies a region on the standard optical diagnostic diagrams[14,74] that is exclusively populated by AGN systems, both at low and high redshift. We note that some of the measured emission lines (notably [O III] and Hβ) trace the outflow rather than the galaxy, which may complicate the interpretation of the line ratios. Nonetheless, the line ratios measured in COSMOS-11142 are fully consistent with those measured in the outflows of high-redshift quasars[25,75].

Despite the rather extreme line ratios, the AGN activity in COSMOS-11142 is relatively weak, leaving no trace other than the ionized gas emission lines. We do not detect AGN emission in the ultraviolet-to-infrared broadband photometry (which is well fit by a model without AGN

contribution), in the mid-infrared colours derived from Spitzer Infrared Array Camera (IRAC) observations (using the criterion proposed by ref. 76), in the rest-frame optical morphology (no evidence for a point source in the NIRCam imaging), in X-ray observations[77] (luminosity $L_X < 10^{44}$ erg s$^{-1}$ at 2–10 keV) or in radio observations[78] (flux $S < 3 \times 10^{23}$ W Hz$^{-1}$ at 3 GHz). We estimate the bolometric luminosity of the AGN from the [O III] luminosity[79], obtaining $\log L_{bol}/(\text{erg s}^{-1}) \approx 45.3$. Using AGN scaling relations[80,81], we find that the upper limit on the radio emission is above the expected level for a typical AGN of this luminosity, whereas the X-ray upper limit is about equal to the value we would expect for a typical AGN of this luminosity. However, the scaling relations present a large scatter, which may be due to the fact that AGN activity is highly variable, and different tracers respond on different timescales[82,83]. We thus conclude that at $z > 2$, current radio and X-ray surveys can only probe the brightest AGNs and are not sensitive to the emission from less extreme systems such as COSMOS-11142. Finally, we point out that the bolometric luminosity we derived from [O III] should be considered an upper limit, as part of the observed line luminosity may come from shock excitation. The actual bolometric luminosity may be an order of magnitude lower; even so, the mass outflow rate that we measure for the ionized phase would not be substantially off from the known AGN scaling relations[10]. Interestingly, the scaling relations would then predict a molecular outflow with a mass rate similar to what we measure for the neutral phase. This suggests that the neutral and molecular phases may be comparable. However, with current data, we are not able to directly probe the molecular gas in COSMOS-11142, and we point out that this system is substantially different from most of the galaxies used to derive the scaling relations, due to its gas-poor and quiescent nature.

## Data availability

The reduced NIRSpec data, the Prospector best fit and the multi-band photometry for COSMOS-11142 are publicly available at https://doi.org/10.5281/zenodo.10058978 (ref. 84).

## Code availability

We used publicly available code including the JWST data reduction pipeline (https://github.com/spacetelescope/jwst), Prospector[39], Forcepho (B.D.J. et al., manuscript in preparation; https://github.com/bd-j/forcepho), dynesty[50], astropy[85], pyneb[61], linetools[86] and specutils[87].

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

**Acknowledgements** We acknowledge discussions with M. Brusa, K. Glazebrook, S. Kulkarni, L. Ciotti, A. Ferrara and A. B. Newman. The Blue Jay Survey is funded in part by STScI Grant JWST-GO-01810. S.B. is supported by the ERC Starting Grant 'Red Cardinal', GA 101076080. R.L.D. is supported by the Australian Research Council Centre of Excellence for All Sky Astrophysics in 3 Dimensions (ASTRO 3D), through project number CE170100013. R.E. acknowledges the support from grant numbers 21-atp21-0077, NSF AST-1816420 and HST-GO-16173.001-A, as well as the Institute for Theory and Computation at the Center for Astrophysics. R.W. acknowledges funding of a Leibniz Junior Research Group (project number J131/2022) This work is based on observations made with the NASA/ESA/CSA James Webb Space Telescope. The data were obtained from the Mikulski Archive for Space Telescopes at the Space Telescope Science Institute, which is operated by the Association of Universities for Research in Astronomy, Inc., under NASA contract NAS 5-03127 for JWST. These observations are associated with programme GO 1810. This work also makes use of observations taken by the 3D-HST Treasury Program (GO 12177 and 12328) with the NASA/ESA HST, which is operated by the Association of Universities for Research in Astronomy, Inc., under NASA contract NAS5-26555.

**Author contributions** S.B. led the observational programme, the analysis and the writing of the paper. M.P. carried out the spectral fitting with Prospector. R.L.D. led the development of the outflow models. J.T.M. reduced the spectroscopic data. B.D.J. carried out the fit of the imaging data. C.C. first identified the neutral outflow in the NIRSpec data. All authors contributed to the planning and execution of the Blue Jay survey, to the interpretation of the results and to the writing of the paper.

**Competing interests** The authors declare no competing interests.

**Additional information**
**Correspondence and requests for materials** should be addressed to Sirio Belli.

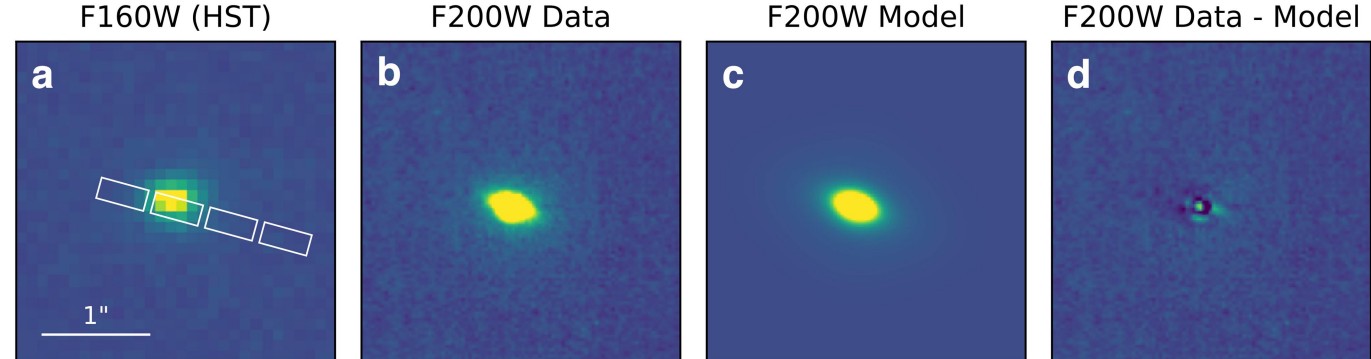

| F160W (HST) | F200W Data | F200W Model | F200W Data - Model |
|---|---|---|---|
| a | b | c | d |

**Extended Data Fig. 1 | Observed and modelled surface brightness distribution. a**, HST F160W data used for designing the observations, with the footprint of the four open MSA microshutters. **b**–**d**, Data, model, and residual for the F200W NIRCam observations. The model is a single Sersic profile obtained with ForcePho, and is able to reproduce the data well.

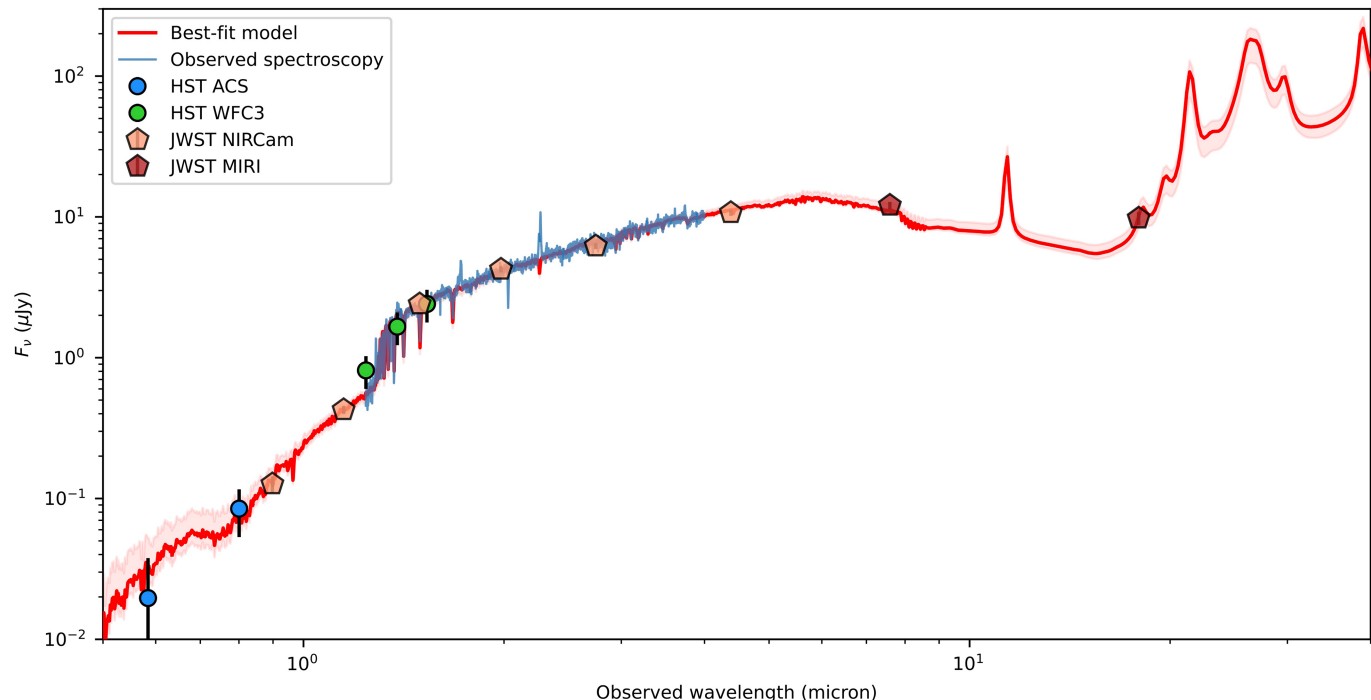

**Extended Data Fig. 2 | Spectral energy distribution.** Points show the observed photometry from the ACS and WFC3 instruments onboard HST (circles), and from the NIRCam and MIRI instruments onboard JWST (pentagons). The observed NIRSpec spectrum is shown in blue, and the red line represents the best-fit model from Prospector, with the shaded red region marking the central 95% credible interval.

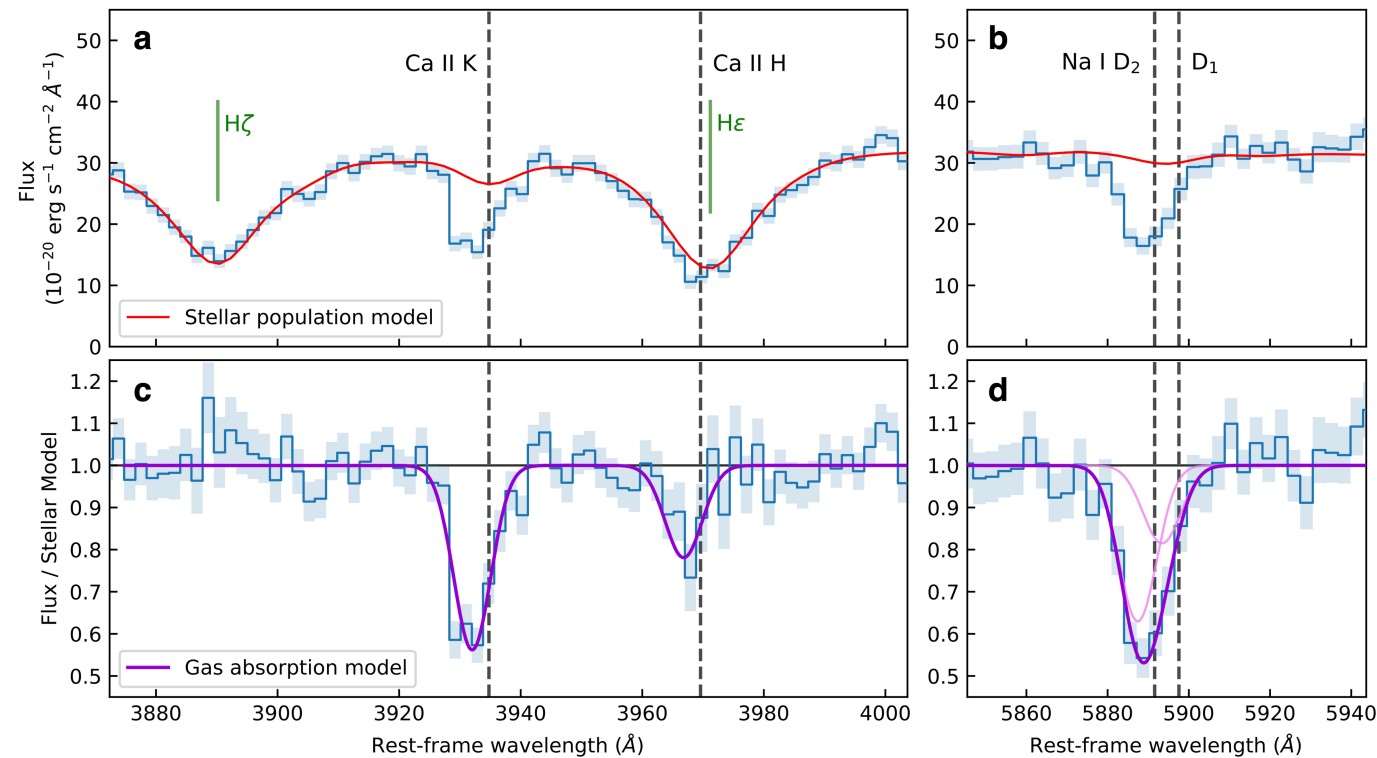

**Extended Data Fig. 3 | Absorption lines from neutral gas. a,b**, Observed spectrum (blue) and best-fit stellar model (red). **c,d**, Ratio of the observed spectrum to the stellar model (blue) and best-fit Gaussian components describing absorption by neutral gas (purple). The expected positions of the resonant lines (at the systemic velocity of the galaxy) are shown in grey, while the positions of the Balmer lines, which are only present in the stellar spectrum, are shown in green. The absorption by neutral gas is clearly blueshifted.

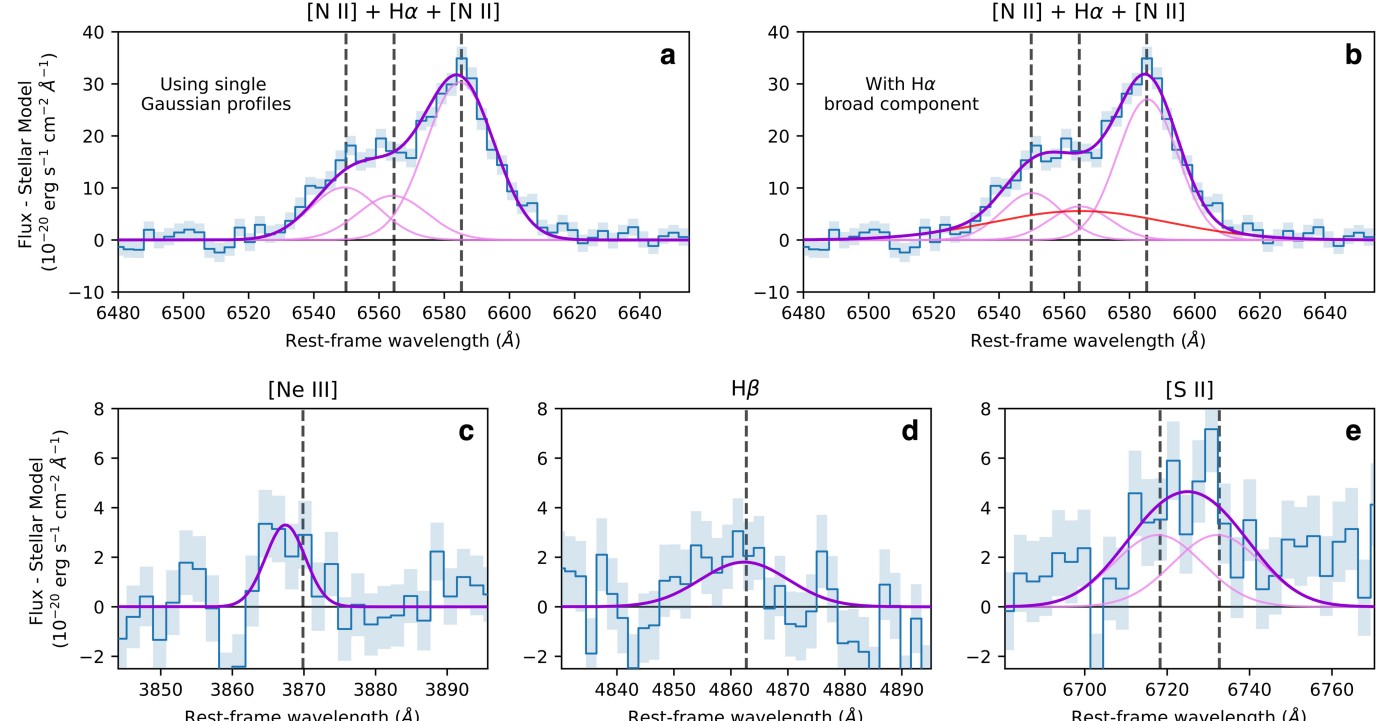

**Extended Data Fig. 4 | Emission lines from ionized gas. a**, Fit to the Hα and [N II] lines, using three single-Gaussian profiles. **b**, Fit to the Hα and [N II] lines, using three single-Gaussian profiles and one additional broad Hα component (shown in red). **c**–**e**, Fit to the [Ne III], Hβ, and [S II] emission lines. In all panels, the spectra shown are obtained by subtracting the best-fit stellar model obtained with Prospector from the observed spectrum. Gaussian fits are shown in purple, with lighter lines showing fits to the individual lines when multiple components are fit simultaneously. The dashed vertical lines mark the expected rest-frame location of the emission lines. Other emission lines are shown in Fig. 2.

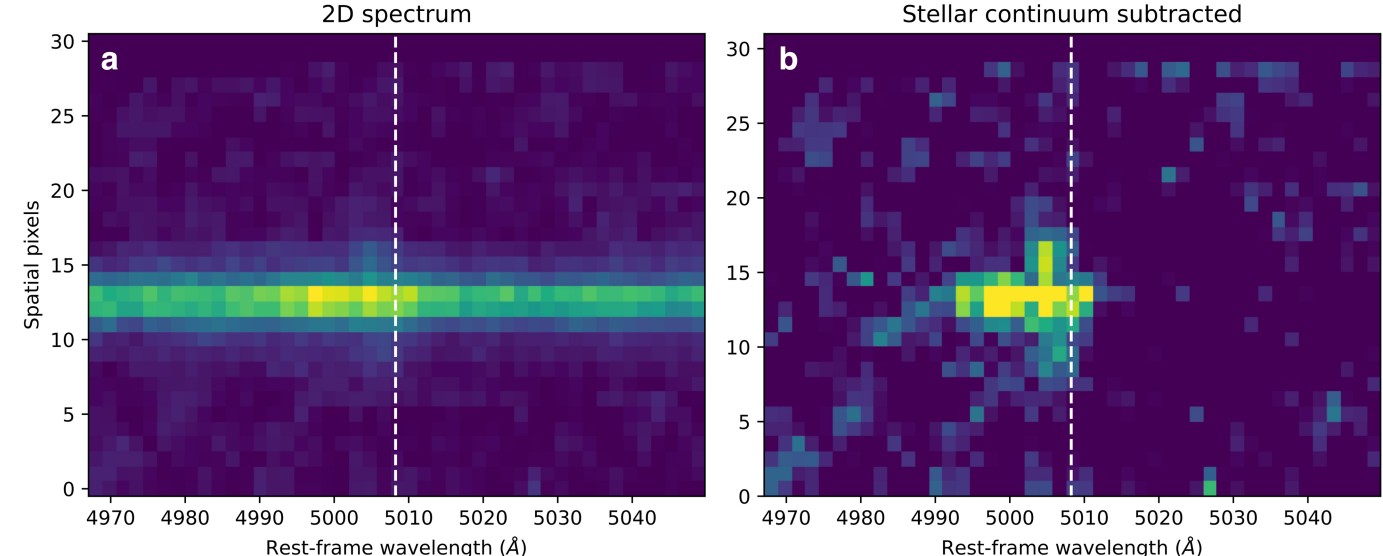

**Extended Data Fig. 5 | Two-dimensional JWST/NIRSpec spectrum centred on the [O III] λ5008 emission line. a**, Observed trace, which is mostly due to stellar emission. **b**, Spatially resolved [O III] emission after subtraction of the median stellar continuum. The vertical dashed line marks the expected position of [O III] at the systemic velocity.

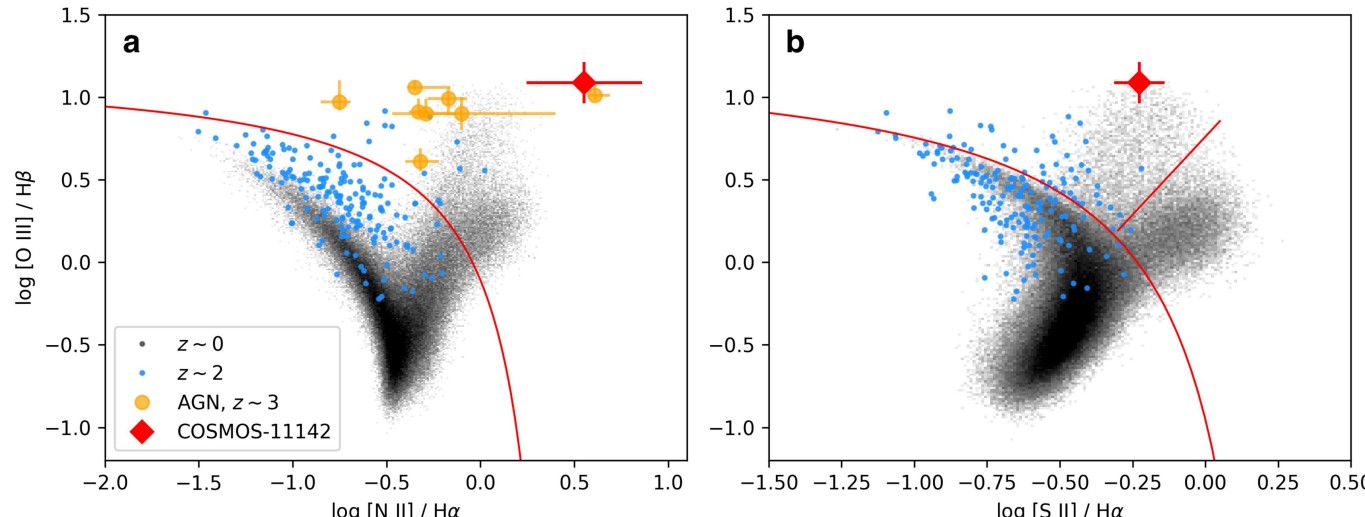

**Extended Data Fig. 6 | Emission line diagnostics. a,b,** Line ratio diagrams[14,74] comparing COSMOS-11142 (red symbol) to a sample of local galaxies (from the Sloan Digital Sky Survey[88], black points), high-redshift galaxies (1.9 < z < 2.7, from the MOSDEF survey[89], blue points) and a sample of high-redshift AGNs observed with JWST/NIRSpec (3 < z < 3.7, from the GA-NIFS survey[90], orange points). The uncertainty on the [N II]/Hα ratio of COSMOS-11142 includes a systematic contribution from the adoption of a specific profile when fitting the emission lines. Red curved lines represent the theoretical maximum starburst locus[91], while the red straight line in panel **b** shows an empirical separation between Seyferts and LINERS[92].

**Extended Data Table 1 | Measured properties of absorption and emission lines**

| Line | $\lambda_{rest}$ (Å) | $\Delta v$ (km/s) | $\sigma$ (km/s) | Flux ($10^{-18}$ erg s$^{-1}$ cm$^{-2}$) | EW (Å) |
|---|---|---|---|---|---|
| **Absorption** | | | | | |
| Ca II K | 3935 | $-210 \pm 19$ | $190 \pm 23$ | | $3.3 \pm 0.2$ |
| Na I D | 5892, 5898 | $-212 \pm 27$ | $187 \pm 41$ | | $6.5 \pm 0.5$ |
| **Emission** | | | | | |
| [O II] | 3727, 3730 | $67 \pm 25$ | $292 \pm 29$ | $4.2 \pm 0.3$ | |
| [Ne III] | 3870 | $-190 \pm 78$ | $168 \pm 99$ | $0.8 \pm 0.2$ | |
| H$\beta$ | 4863 | $-22^{\dagger}$ | $465^{\dagger}$ | $1.2 \pm 0.3$ | |
| [O III] | 5008 | $-438 \pm 30$ | $632 \pm 31$ | $14.3 \pm 0.6$ | |
| H$\alpha$ | 6565 | $-22 \pm 23$ | $465 \pm 23$ | $7.8 \pm 1.1$ | |
| [N II] | 6585 | $-22 \pm 23$ | $465 \pm 23$ | $27.7 \pm 1.1$ | |
| [S II] | 6718, 6733 | $-22^{\dagger}$ | $465^{\dagger}$ | $5.4 \pm 0.7$ | |
| [S III] | 9533 | $-61 \pm 131$ | $591 \pm 133$ | $5.5 \pm 1.0$ | |
| He I | 10833 | $386 \pm 79$ | $531 \pm 81$ | $7.4 \pm 0.9$ | |

Quantities marked with † have been fixed during the fit. The kinematics for Hα and [N II] are assumed to be identical.

**Extended Data Table 2 | Outflow properties measured from different lines**

| | $v_{out}$ <br> ($10^3$ km/s) | $M_{out}$ <br> ($10^6 M_\odot$) | $\dot{M}_{out}$ <br> ($M_\odot$/yr) |
|---|---|---|---|
| **Neutral outflow** | | | |
| Na I D | 0.21 | 488 | 35 |
| Ca II K | 0.21 | > 9 | > 0.65 |
| **Ionized outflow** | | | |
| [Ne III] | 0.36 | 1.44 | 0.18 |
| H$\beta$ | 0.93 | 1.16 | 0.37 |
| [O III] | 1.07 | 1.98 | 0.72 |
| [S III] | 1.18 | 2.46 | 0.99 |