## [Peer Review File · Nature]

Manuscript Title: Star Formation Shut Down by Multiphase Gas Outflow in a Galaxy at $z = 2.45$

Reviewer Comments & Author Rebuttals

Reviewer Reports on the Initial Version:

Referees' comments:

Referee #1 (Remarks to the Author):

Overall, I recommend the publication of this article in Nature or Nature Astronomy. A detailed assessment of this article is given below following the points mentioned in the referee form.

A. Summary of the key results

This paper presents the first convincing case to date of a massive AGN-driven outflow in a massive gas-poor galaxy that is in the process of quenching its star formation activity at cosmic noon (redshifts $z \sim 2-3$). Most numerical simulations of galaxy formation and evolution predict that this process of "ejective AGN feedback" is important in massive galaxies at that epoch, yet observations of this phenomenon is still very limited. Previous cases of massive outflows like this one have been detected in gas-rich starburst and active galaxies, near and far, but none in gas-poor, quiescent galaxies that lie below the main sequence of star-forming galaxies. The authors argue rather convincingly that this outflow is mostly neutral-atomic rather than warm-ionized, and that it is massive enough to explain the rapid suppression of star formation in this system.

B. Originality and significance: if not novel, please include reference

To my knowledge, this is the first detection of a massive AGN-driven outflow in a quenching gas-poor galaxy at cosmic noon, despite significant efforts and investments of telescope time in the past. This high-S/N detection of a powerful cool-gas outflow at $z \sim 2.4$, seen in absorption, is made possible by the excellent sensitivity and angular resolution of JWST in the near-infrared.

Most modern simulations of galaxy formation and evolution require some form of "ejective feedback" to reproduce the universe at redshift $z = 0$, particularly at the high-mass end of the galaxy function. Its detection in one galaxy is an important first step to demonstrate that it indeed plays a role in shaping massive galaxies in the universe. These results shed new light on galaxy formation and evolution, one of the key open questions in extragalactic astronomy today. These results will have an impact on people working in this field of research and on those outside the field who have a general interest to know how the current universe was put together.

C. Data & methodology: validity of approach, quality of data, quality of presentation

The rest-frame optical - near-infrared spectrum used by the authors for their spectroscopic analysis is of exquisite quality, comparable in quality to those of the best-studied local galaxies. The authors are therefore able to apply time-proven methods of analysis of the absorption and emission line features to derive robust numbers on the host and outflow properties and come to convincing, carefully worded conclusions.

D. Appropriate use of statistics and treatment of uncertainties

Except for the missing uncertainty on the stellar mass (noted below under "Suggested improvements"), the authors' use of statistics and treatment of uncertainties is on par with some of the best measurements of host galaxy and outflow properties in the nearby universe. In particular, the authors are honest about the rather large uncertainties on the mass outflow rates.

E. Conclusions: robustness, validity, reliability

I find the conclusions to be quite convincing given their conservative treatment of the uncertainties on the outflow measurements.

F. Suggested improvements: experiments, data for possible revision

Following the order of appearance in the paper:

p. 3. The authors use $v_{\text{out}} = |\Delta v| + 2 \sigma$, but this expression refers to the maximum outflow velocity, not the "typical" (median) velocity of the outflowing gas. As a consequence, in my opinion, dM/dt is overestimated. The authors should investigate the impact of choosing $|\Delta v|$ and/or $|\Delta v| + 1 \sigma$, instead of $|\Delta v| + 2 \sigma$, as the characteristic outflow velocity on their estimates of dM/dt (and change their conclusions accordingly, if needed).

pp. 5 and 14. The ratio between molecular, neutral-atomic, and warm-ionized mass outflow rates vary widely among local outflows. The ratio of (molecular + neutral-atomic) to (warm-ionized) mass outflow rates is typically large in gas-rich systems (e.g. ULIRGs and dusty quasars; Rupke, Gultekin, & Veilleux 2017), but this ratio is likely lower in gas-poor systems, such as COSMOS-11142, due to a general deficit of cool gas in these systems. I therefore find the statement "the molecular phase (not probed by our observations) is expected to be equally effective as the neutral phase at ejecting gas" to be rather optimistic given that COSMOS-11142 is gas-poor and with an unknown amount of molecular gas. I suggest to tone it down. Similarly, the assumption on p. 14 that "the neutral phase is comparable to the molecular phase" is fraught with large uncertainties.

pp. 6 and 14. The lack of X-ray and radio detection of the AGN in this object is somewhat intriguing given

the inferred (upper limit on the) AGN bolometric luminosity, $\log L_{\text{bol}} \sim 45.3$. Possible explanations should be briefly mentioned.

pp. 7, 8, and ff. As shown in Fig. 4, the MSA microshutters of NIRSpec miss a significant fraction of the galaxy, including part of the nucleus. The authors estimate on p. 8 that the slit-loss correction is ~ 2 . However, this estimate is based on photometric measurements. I worry that this correction could be more severe when considering quantities that are derived from the absorption and emission line features, such as the outflow properties (velocity, column, mass, etc), since the line emission and absorption associated with the outflow may not be homogeneous and distributed spherically symmetrically around the nucleus. The authors should address this issue and possibly add it in their measurement uncertainties.

p. 8. The stellar mass, $\log M^*/M_{\text{sun}} = 10.9$, reported on this page is missing an uncertainty. This makes it hard to judge whether the discrepancy discussed later on pp. 8-9 between dynamical and stellar masses is significant. The authors should quantify this discrepancy given the uncertainties (e.g. "...different at the N-sigma level" ...)

p. 10. To explain that H α appears to be slightly blueshifted compared to [N II], the authors speculate that perhaps "H α mostly comes from the outflow while [N II] originates in the galaxy". This statement is hard to justify given that both features trace warm ionized gas, unless the excitation, density, or metallicity is also different between these two regions. The authors should clarify what they have in mind here.

p. 13. The authors should be more explicit and state that the "canonical value" used for dust depletion refers to the Milky Way value and let the readers judge if it is a reasonable assumption here.

p. 14 and Fig. 9 caption. The authors should replace all statements about the "Baldwin-Phillips-Terlevich (BPT) diagrams" with the "standard optical diagnostic diagrams of BPT and Veilleux & Osterbrock (1987)" since the [S II] diagram they use for their analysis was first introduced in 1987 by Veilleux & Osterbrock (the spectra of BPT did not extend far enough to the red to include the [S II] doublet).

G. References: appropriate credit to previous work?

Yes, overall, the authors give appropriate credit to previous work.

H. Clarity and context: lucidity of abstract/summary, appropriateness of abstract, introduction and conclusions

The overall clarity of the abstract, introduction, and conclusion is excellent.

Referee #2 (Remarks to the Author):

The manuscript presents ~ 1000 JWST NIRSpec data of a $z \sim 2.5$ galaxy at rest-frame wavelengths $\sim 3700 \text{ \AA}$ to $\sim 1.1 \text{ \mu m}$. Based on photometry and spectroscopy, this galaxy is quiescent, with strong star formation in the somewhat recent past ($\sim 200 \text{ Myr}$ ago). The spectra shows evidence for a neutral gas outflow seen in absorption, as well as an ionized outflow seen in emission.

This detection is unique in a couple of ways. First, it extends the technique of using rest-frame optical interstellar absorption lines to detect outflows to much higher redshift. Second, it shows an outflow in a massive galaxy at cosmic noon without strong ongoing star formation, but possible AGN activity.

While these data showcase the possibility of JWST for this work, I'm not convinced that it should be published in Nature. The science is interesting within the niche of establishing neutral gas outflow properties in post-starburst galaxies, but its broader appeal would be limited. Furthermore, the data itself has limitations: the result is primarily based on an optical spectrum of relatively low spectral and spatial resolution. The analysis of this spectrum needs improvement, but I'm not convinced the data will allow the improvements required to address uncertainties in the line fitting. Better data is available from, e.g., the NIRSpec IFU mode of JWST with higher-resolution gratings. A paper on a very similar galaxy to this one, in fact, has just been submitted to arXiv and Nature Astronomy, but with the benefit of IFU data (arXiv:2308.06317). I thus suggest that Nature Astronomy or ApJ Letters would be a more appropriate venue for this result.

In terms of analysis and interpretation, I have the following major concerns:

1. The line fitting is not presented in enough detail to assess its quality. Each line is fit separately, but the signal-to-noise and spectral resolution are low so it is hard to assess whether the fits to individual lines are really robust, even as presented in Figures 2, 6, and 7. Certainly at higher resolution some of these lines would break up into multiple components, and even now the data seem to point to a second blueshifted component in [OII] and [OIII], at least judging from Figures 1 and 8. Some of the fits are not correct, such

as near [SII] where it appears the stellar model underfits the continuum, leading to more flux in the emission lines. A better correction needs to be applied here. There is no limit on at least one important diagnostic line, [OI] 6300 Å, typically used in line ratio diagrams to assess ionization and excitation. I notice lines present that are not fit, such as the [NI] 5198/5200 doublet. In general, I am skeptical that the emission lines should show so much differences from each other in kinematics; e.g., that H α should be blueshifted by so much.

The authors should at least attempt a global fit of all lines, or at least define 2 or 3 groups of lines to fit together with similar kinematics. This would provide better constraints on the weak lines, but also on strong lines with degeneracies (e.g., H α due to the strength of [NII]).

I am consequently concerned about the estimate of \dot{M} from different species. That assumes a lot about the gas ionization and other uncertainties in different species. It is most reliable to use Balmer lines, which don't require such corrections (except for density).

2. Dust. The interpretation of the emission lines, particularly [SIII] and HeI, given in Figure 1 and the text is based primarily on arguments around dust obscuration. For this to be robust, it would need to be quantified using a wavelength-dependent extinction curve. Furthermore, there needs to be discussion of the gas extinction based on observed ionized emission lines. The authors currently seem to rely on extrapolation from stellar extinction in the SED model (which I assume includes the usual correction from stellar to gas color excess; Calzetti et al. 2000).

The fits to [NII] and [SIII] seem very similar, so I don't think [SIII] has its shape due to differential dust attenuation. There are two [SIII] lines -- do they yield the same result?

The HeI is interesting, and I like the interpretation -- have the authors double-checked for wavelength calibration issues?

3. Velocities. The manuscript uses more-or-less a "maximum" velocity as the bulk velocity of the gas. But most of the gas, as detected, is at the central velocity of each line. This is what is appropriate for calculating mass outflow rates. There may be some unresolved

projection effects, though not in the model as presented. Thus the outflow rates will be lower by a factor of a few.

4. Size. It is not obvious from Figure 8 that the highly-blueshifted [OIII] line is spatially-resolved. The size the manuscript suggests appears to use the size of the less blueshifted part of the line. This circles back to the point about line fitting above -- are there constraints on the spatial extent of the other observed lines? Are there two outflows here, one lower-velocity, spatially resolved and one high-velocity, unresolved?

5. Presence of an AGN. The presence of an AGN is inferred from the galaxy's position on the line ratio diagrams in Figure 9. However, the only contextualization is comparison to low-z galaxies, for which this system appears as an extreme outlier. A robust comparison to recent results on both high-z star-forming galaxies and AGN is warranted. Galaxies shift on these diagrams with increasing redshift. Simply comparing to low-z data and models is insufficient.

One indicative point in favor of an AGN is the combination of the strong blueshift of [OIII] with the high-velocity component being unresolved.

6. Covering fraction. The covering fraction can in fact be determined from the existing data; see, e.g., the formulae in Section 4.3 of Hamann et al. 1997, ApJ, 478, 80.

7. Absorption line fitting: The model for Call shown in Figure 6 does not look like a 2-to-1 optical depth ratio -- it would be better to fix it to 2, or at the very least set that as an upper limit for the ratio, since it physically can't be larger than that in the optically-thin limit.

The authors assume that NaI is optically thin based on Call. However, NaI is typically mildly optically thick when observed in these contexts. Call H and K are not usually observed, so the fact that the gas is dense enough to detect it suggests that NaI is even more likely to be optically thick. (The difference in detection rate presumably arises due to the large difference in dust depletion, given that they have similar oscillator strengths.) A fit with the line ratio allowed to vary would be more appropriate. If in fact the optical depth is higher, the covering factor may differ from unity.

The optical depth and blueshift of NaI D may also be a lower and upper limit, respectively, if in fact there is redshifted resonant emission that is filling in the red side of the absorption line. The presence of redshifted HeI could suggest that such emission is a real possibility.

Minor textual point:

- Please quantify the statement "relatively weak emission lines" as a indicator of quiescence. At issue is really the equivalent width of the Balmer emission lines. Similarly, the manuscript states that the "deep Balmer absorption lines clearly indicates that the galaxy is in the `post-starburst` phase." Again, the indicator is this combined with low Balmer emission line equivalent width, quantitatively.

Author Rebuttals to Initial Comments:

We thank both referees for their thorough feedback, which helped improve the quality of the revised manuscript. We start by discussing two important points that were raised by both referees, and continue by responding in detail to each referee report. We mark all changes in the manuscript, including those that were not requested by the referees but were necessary to meet the word limit of the main part and/or to conform to the journal format. We also corrected small imprecisions, and updated the uncertainties of the emission lines after running the fits with updated software (the changes are insignificant and only apply to the uncertainties, not the measurements).

a) Outflow velocity

Both referees point out that our definition of outflow velocity as $v_{\text{out}} = |\Delta v| + 2\sigma$ is an overestimate of the bulk velocity of the gas in the outflow. We chose this definition because it accounts for unresolved projection effects; however we now realize that this choice is not suitable for our model of the neutral outflow (while it is still appropriate for the ionized outflow). To make our model clearer we have revised Figure 2 in the paper, where we now show the different location of the neutral and ionized phases in the outflow. The revised figure also makes it clear that the outflow should not be understood as a relatively narrow beam directed towards the observer: this is both unlikely and inconsistent with the high incidence of detected neutral outflows in our Blue Jay survey (R. L. Davies et al. 2023). The opening angle could be as wide as 4π , in which case the outflow is spherical and not biconical; the qualitative discussion of the outflow geometry remains the same.

The **ionized phase** of the outflow is modeled as an expanding bicone (or sphere), and therefore the observed velocity distribution reflects a wide range of angles between the gas velocity and the line of sight. In this case the intrinsic outflow velocity corresponds to the largest observed amount, and so for the ionized outflow we adopt $v_{\text{out}} = |\Delta v| + 2\sigma$. We also point out that, under the assumption of a mass-conserving outflow, the gas density in the bicone scales with radius as r^{-2} . In this scenario, most of the ionized emission must come from the central regions, because the line emission luminosity scales as $n^2 \sim r^{-4}$. This is confirmed by the fact that we see most of the [O III] emission coming from small galactocentric radii; and explains why, despite the large size of the bicone, the dust in the galaxy is able to absorb a substantial amount of light for some of the emission lines. Anyway, if we were to choose $v_{\text{out}} = |\Delta v|$ the mass outflow rate for the ionized phase would be smaller and our main conclusions would not change.

The **neutral phase** of the outflow is likely to be farther out from the galaxy compared to the ionized phase in order to remain neutral, and the simplest model is that of a thin shell. Since we clearly see ionized emission out to ~ 3 kpc, we conclude that the neutral shell has a large size compared to the galaxy effective size (0.6 kpc). For this reason, the line of sight to the stellar light intercepts only a small part of the neutral outflow, and in this region the gas is directed exactly at the viewer, as illustrated in the updated Figure 2. In this scenario, the observed line-of-sight velocity distribution is not due to a

range of angles, but reflects the range of intrinsic velocities. Thus, we take the central velocity as representative of the bulk outflow velocity: $v_{\text{out}} = |\Delta v|$. As a consequence of this change, in the revised manuscript the outflow velocity of the neutral phase is ~ 3 times smaller, and the outflow mass rate is also ~ 3 times smaller. This does not change the main conclusions of the study, since the neutral outflow is still sufficiently powerful to quench the galaxy. We also note that in the alternative “conservative” case in which the neutral outflow is much smaller, with $R_{\text{out}} \sim R_{\text{eff}}$, the unresolved projection effects become important for the neutral phase (since now we would be able to probe the full outflow in absorption against the stellar emission, and not just a small patch coming towards us). In this case we keep the old definition $v_{\text{out}} = |\Delta v| + 2\sigma$, and the lower limit on the mass outflow rate of $11 M_{\text{sun}}/\text{yr}$ remains unchanged.

b) Fit to the Halpha emission

Following the remarks made by both referees, we have changed the treatment of Halpha in the emission line fits. We were previously fixing the Halpha kinematics to those of Hbeta, but this was not a good idea for at least two reasons: Hbeta is noisy and may not provide the best estimate of the velocity; and differential extinction may produce substantially different line profiles for Hbeta and Halpha. Instead, we now tie the kinematics of Halpha to that of [N II]. Since the three lines are blended, this fit is highly degenerate, and to illustrate this we have added a panel to Figure 7 where we show the result of adding a broad component for Halpha, which gives a marginally better fit to the data. The large velocity dispersion of the broad component ($\sigma_{\text{broad}} \sim 1200 \text{ km/s}$) would be consistent with the presence of a weak Broad Line Region; however, we obtain a virtually identical fit if we add the broad component to the [N II] line as well ($\sigma_{\text{broad}} \sim 800 \text{ km/s}$), which could be interpreted as part of the outflow seen in low-ionization lines. This exercise shows that the lines are too blended to extract reliable kinematics. In our analysis we only use the Halpha flux as an upper limit to the star formation, and the [N II]/Halpha ratio for the line diagnostic diagrams; in both cases our results are qualitatively unchanged if we adopt different models for the emission lines.

Referee #1 (Remarks to the Author):

Overall, I recommend the publication of this article in Nature or Nature Astronomy. A detailed assessment of this article is given below following the points mentioned in the referee form.

A. Summary of the key results

This paper presents the first convincing case to date of a massive AGN-driven outflow in a massive gas-poor galaxy that is in the process of quenching its star formation activity at cosmic noon (redshifts $z \sim 2-3$). Most numerical simulations of galaxy formation and evolution predict that this process of "ejective AGN feedback" is important in massive galaxies at that epoch, yet observations of this phenomenon is still very limited. Previous cases of massive outflows like this one have been detected in gas-rich starburst and active galaxies, near and far, but none in gas-poor, quiescent galaxies that lie below the main sequence of star-forming galaxies. The authors argue rather convincingly that this outflow is mostly neutral-atomic rather than warm-ionized, and that it is massive enough to explain the rapid suppression of star formation in this system.

B. Originality and significance: if not novel, please include reference

To my knowledge, this is the first detection of a massive AGN-driven outflow in a quenching gas-poor galaxy at cosmic noon, despite significant efforts and investments of telescope time in the past. This high-S/N detection of a powerful cool-gas outflow at $z \sim 2.4$, seen in absorption, is made possible by the excellent sensitivity and angular resolution of JWST in the near-infrared.

Most modern simulations of galaxy formation and evolution require some form of "ejective feedback" to reproduce the universe at redshift $z = 0$, particularly at the high-mass end of the galaxy function. Its detection in one galaxy is an important first step to demonstrate that it indeed plays a role in shaping massive galaxies in the universe. These results shed new light on galaxy formation and evolution, one of the key open questions in extragalactic astronomy today. These results will have an impact on people working in this field of research and on those outside the field who have a general interest to know how the current universe was put together.

C. Data & methodology: validity of approach, quality of data, quality of presentation

The rest-frame optical - near-infrared spectrum used by the authors for their spectroscopic analysis is of exquisite quality, comparable in quality to those of the best-studied local galaxies. The authors are therefore able to apply time-proven methods of analysis of the absorption and emission line features to derive robust numbers on the host and outflow properties and come to convincing, carefully worded conclusions.

D. Appropriate use of statistics and treatment of uncertainties

Except for the missing uncertainty on the stellar mass (noted below under "Suggested improvements"), the authors' use of statistics and treatment of uncertainties is on par with some of the best measurements of host galaxy and outflow properties in the nearby universe. In particular, the authors are honest about the rather large uncertainties on the mass outflow rates.

E. Conclusions: robustness, validity, reliability

I find the conclusions to be quite convincing given their conservative treatment of the uncertainties on the outflow measurements.

F. Suggested improvements: experiments, data for possible revision

Following the order of appearance in the paper:

p. 3. The authors use $v_{\text{out}} = \Delta v + 2\sigma$, but this expression refers to the maximum outflow velocity, not the "typical" (median) velocity of the outflowing gas. As a consequence, in my opinion, dM/dt is overestimated. The authors should investigate the impact of choosing Δv and/or $\Delta v + 1\sigma$, instead of $\Delta v + 2\sigma$, as the characteristic outflow velocity on their estimates of dM/dt (and change their conclusions accordingly, if needed).

For a detailed answer, see point a) at the beginning of this document.

pp. 5 and 14. The ratio between molecular, neutral-atomic, and warm-ionized mass outflow rates vary widely among local outflows. The ratio of (molecular + neutral-atomic) to (warm-ionized) mass outflow rates is typically large in gas-rich systems (e.g. ULIRGs and dusty quasars; Rupke, Gultekin, & Veilleux 2017), but this ratio is likely lower in gas-poor systems, such as COSMOS-11142, due to a general deficit of cool gas in these systems. I therefore find the statement "the molecular phase (not probed by our observations) is expected to be equally effective as the neutral phase at ejecting gas" to be rather optimistic given that COSMOS-11142 is gas-poor and with an unknown amount of molecular gas. I suggest to tone it down. Similarly, the assumption on p. 14 that "the neutral phase is comparable to the molecular phase" is fraught with large uncertainties.

The referee makes a good point; we have removed the first statement and revised the second one.

pp. 6 and 14. The lack of X-ray and radio detection of the AGN in this object is somewhat intriguing given the inferred (upper limit on the) AGN bolometric luminosity, $\log L_{\text{bol}} \sim 45.3$. Possible explanations should be briefly mentioned.

We expanded on the X-ray and radio non-detections in the last section of the Methods. We now quote the upper limits on the X-ray and radio flux, and compare to the AGN scaling relations.

pp. 7, 8, and ff. As shown in Fig. 4, the MSA microshutters of NIRSpec miss a significant fraction of the galaxy, including part of the nucleus. The authors estimate on p. 8 that the slit-loss correction is ~ 2 . However, this estimate is based on photometric measurements. I worry that this correction could be more severe when considering quantities that are derived from the absorption and emission line features, such as the outflow properties (velocity, column, mass, etc), since the line emission and absorption associated with the outflow may not be homogeneous and distributed spherically symmetrically around the nucleus. The authors should address this issue and possibly add it in their measurement uncertainties.

Given the observed complexity of the [O III] spatial distribution and its large spatial extent, it is possible that the slit correction for the emission lines is different from that for the stellar continuum. We have added a brief discussion of this issue. For the neutral gas absorption lines this effect is not important, because the amount of gas that does not overlap with the stellar emission is already included in the calculation (and uncertainty estimate) through the opening angle Ω .

p. 8. The stellar mass, $\log M^*/M_{\text{sun}} = 10.9$, reported on this page is missing an uncertainty. This makes it hard to judge whether the discrepancy discussed later on pp. 8-9 between dynamical and stellar masses is significant. The authors should quantify this discrepancy given the uncertainties (e.g. "...different at the N-sigma level"...)

We have removed the comparison of dynamical and stellar mass for the following reason. Very recently, de Graaff et al. (arXiv:2308.09742) showed that the NIRSpec spectral resolution for point sources is up to 1.8 times higher than the nominal value (which is calculated for a source that fills up the slit). Given the small size of our target, it is possible that the true instrumental resolution of our data is better than the nominal value, and therefore all our velocity dispersion measurements are slightly underestimated. For the stellar kinematics, we measure $\sigma_{\text{star}} = 273$ km/s and assume $\sigma_{\text{instr}} \sim 130$ km/s (approximately; this is actually a function of wavelength). If the real instrumental resolution were 1.8 times better than nominal, i.e. $\sigma_{\text{instr}} \sim 70$ km/s, the observed spectrum would yield a measured velocity dispersion of $\sigma_{\text{star}} = 294$ km/s. The corresponding dynamical mass would then be very close to the stellar mass. This exercise convinced us that we cannot make a statement on the stellar vs. dynamical mass comparison without properly taking into account all the systematics involved, which we defer to a future work.

We note that the instrumental resolution issue affects the velocity dispersion measurement for the gas emission and absorption lines as well, but is too small to affect the derived outflow properties in any meaningful way.

Now that we removed the quantitative comparison from the article, and we quote the stellar mass solely to give some context, we prefer to leave the stellar mass without uncertainty. The formal uncertainty from the spectral fitting is small, 0.03 dex, and would

give a false impression of accuracy. The real uncertainty is mostly systemic, and is larger than 0.2 dex (e.g. Mobasher et al. 2015), but difficult to quantify.

p. 10. To explain that H α appears to be slightly blueshifted compared to [N II], the authors speculate that perhaps "H α mostly comes from the outflow while [N II] originates in the galaxy". This statement is hard to justify given that both features trace warm ionized gas, unless the excitation, density, or metallicity is also different between these two regions. The authors should clarify what they have in mind here.

We now set the H α kinematics to follow those of [N II], as explained in detail in point b) at the beginning of this document.

p. 13. The authors should be more explicit and state that the "canonical value" used for dust depletion refers to the Milky Way value and let the readers judge if it is a reasonable assumption here.

We updated the text specifying that the assumed value refers to the Milky Way.

p. 14 and Fig. 9 caption. The authors should replace all statements about the "Baldwin-Phillips-Terlevich (BPT) diagrams" with the "standard optical diagnostic diagrams of BPT and Veilleux & Osterbrock (1987)" since the [S II] diagram they use for their analysis was first introduced in 1987 by Veilleux & Osterbrock (the spectra of BPT did not extend far enough to the red to include the [S II] doublet).

We thank the referee for spotting this imprecision. We have corrected the references.

G. References: appropriate credit to previous work?

Yes, overall, the authors give appropriate credit to previous work.

H. Clarity and context: lucidity of abstract/summary, appropriateness of abstract, introduction and conclusions

The overall clarity of the abstract, introduction, and conclusion is excellent.

Referee #2 (Remarks to the Author):

The manuscript presents $R \sim 1000$ JWST NIRSpec data of a $z \sim 2.5$ galaxy

at rest-frame wavelengths $\sim 3700 \text{ \AA}$ to $\sim 1.1 \text{ \mu m}$. Based on photometry and spectroscopy, this galaxy is quiescent, with strong star formation in the somewhat recent past ($\sim 200 \text{ Myr}$ ago). The spectra shows evidence for a neutral gas outflow seen in absorption, as well as an ionized outflow seen in emission.

This detection is unique in a couple of ways. First, it extends the technique of using rest-frame optical interstellar absorption lines to detect outflows to much higher redshift. Second, it shows an outflow in a massive galaxy at cosmic noon without strong ongoing star formation, but possible AGN activity.

While these data showcase the possibility of JWST for this work, I'm not convinced that it should be published in Nature. The science is interesting within the niche of establishing neutral gas outflow properties in post-starburst galaxies, but its broader appeal would be limited. Furthermore, the data itself has limitations: the result is primarily based on an optical spectrum of relatively low spectral and spatial resolution. The analysis of this spectrum needs improvement, but I'm not convinced the data will allow the improvements required to address uncertainties in the line fitting. Better data is available from, e.g., the NIRSpec IFU mode of JWST with higher-resolution gratings. A paper on a very similar galaxy to this one, in fact, has just been submitted to arXiv and Nature Astronomy, but with the benefit of IFU data (arXiv:2308.06317). I thus suggest that Nature Astronomy or ApJ Letters would be a more appropriate venue for this result.

One of our results is to establish the presence of a neutral gas outflow in a post-starburst galaxy; and we agree that this could be called a “niche” interest (although we note that post-starburst galaxies are a rare population in the local universe, but represent the main way to form a quiescent galaxy at high redshift). However, the main result of our work is determining that such neutral outflow is capable of quenching the galaxy; this conclusion comes from the joint analysis of the outflow properties and the star formation history derived from spectral fitting. *This* is the result that we believe would interest the broader community.

While most (though not all) experts in the field believe that massive galaxies are quenched by AGN feedback, this hypothesis has not been directly confirmed by observations (for a detailed review see Harrison 2017, Nat As 1, 165). Moreover, it is not clear how AGN feedback impacts star formation in the galaxy: is it by ejecting cold gas or by heating up the gas in the halo and preventing further cooling? After decades of observational efforts, virtually all evidence of AGN feedback comes from studies of gas-rich, star-forming galaxies. The lack of evidence for effective AGN feedback in the galaxy population that is supposedly the most affected by it -- i.e. galaxies that already started the quenching process -- was recognized as one of the major issues in the field

as recently as a few months ago, in the Nature Astronomy report about a recent quenching conference (Curtis-Lake et al. 2023): "So the picture is not wholly clear. It seems that AGNs can power outflows that can conceivably trigger quiescence, but they still reside in galaxies with large reservoirs of molecular gas, the fuel for star formation. However, this line of research does not show a direct causal link between AGN feedback, gas removal (via ejection) and subsequent suppression of star formation." We believe that our observations provide exactly this link between AGN feedback, ejection, and quenching.

The broader impact of our observations is thus two-fold: 1) we provide the first direct evidence of AGN feedback causing quenching at high redshift, and 2) we also show that such feedback has the specific form of gas ejection, as opposed to gas heating. Our interpretation is shared by Referee #1, who states that our results will have an impact "on those outside the field who have a general interest to know how the current universe was put together". To emphasize the main result of our study we have revised the title, which now explicitly states that quenching is due to the multiphase outflow.

As for the analysis of the spectrum, we agree with most of the referee's comments (to which we respond in detail below), however none of these sources of uncertainty affect the main result of our study. Better data would certainly help, but our goal is not to study in detail the velocity structure of the outflow in different phases. The main question we are trying to answer is whether the observed outflow can cause the observed quenching; and the answer is affirmative even if we account for all the issues raised by the referee. Most of the comments about line fitting concern the ionized phase, which provides a negligible contribution to quenching compared to the neutral phase. Finally, while judging the "quality of the data" is unavoidably subjective, we point out that the SNR of our spectrum is about an order of magnitude better than what available pre-JWST, and enables *for the first time* the detection of the [Ne III], Hbeta, [S II], [S III], and He I emission lines in a quiescent galaxy at high redshift.

In terms of analysis and interpretation, I have the following major concerns:

1. The line fitting is not presented in enough detail to assess its quality. Each line is fit separately, but the signal-to-noise and spectral resolution are low so it is hard to assess whether the fits to individual lines are really robust, even as presented in Figures 2, 6, and 7. Certainly at higher resolution some of these lines would break up into multiple components, and even now the data seem to point to a second blueshifted component in [OII] and [OIII], at least judging from Figures 1 and 8.

We agree that the emission lines are likely not consistent with a single Gaussian profile; this can be seen by the discrepancy between the best-fit Gaussian model and the data in some of the brightest lines, particularly [O III]. However, a detailed decomposition of

the observed gas kinematics is beyond the scope of this work. Our goal is to derive a measurement of the flux and velocity shift for each line in order to estimate the mass outflow rate. Given that this estimate is ultimately limited by systematic uncertainties that are very large (especially the contribution from the electron number density), the exact way in which we measure the line fluxes and kinematics has no impact on our results. To test this, we also employed non-parametric measurements of the flux and kinematics, obtaining consistent results. The problem with the non-parametric measurements is that they can only be applied to isolated (non-blended) lines, while using Gaussian profiles allows us to fit cases where two or three lines are not well separated. Thus, the single Gaussian profile model we employ should not be considered as a realistic model of the line profile, but as a tool to measure the line properties. We have added a clarification of this point at the end of the section “Absorption and Emission Lines”.

Some of the fits are not correct, such as near [SII] where it appears the stellar model underfits the continuum, leading to more flux in the emission lines. A better correction needs to be applied here. There is no limit on at least one important diagnostic line, [OI] 6300 Å, typically used in line ratio diagrams to assess ionization and excitation. I notice lines present that are not fit, such as the [NI] 5198/5200 doublet.

The 3-sigma upper limit on the [O I] flux is 1.5×10^{-18} erg/(s cm²), which is only a few times fainter than H α . In order to discriminate between different sources of ionization we would need a ratio of 1/10 or lower (see, e.g., review by Kewley et al. 2019).

Some parts of the spectrum show small residuals from the best fit continuum, such as around 5200Å, but the 2-D morphology of the feature and its wavelength are not consistent with it being associated to [N I] emission. These residuals are likely due to the NIRSPEC data reduction pipeline, which is still being perfected. The same can be said about the residuals around [S II]: we do not think they are due to issues with the stellar model. We note that a much larger residual is present around ~4200Å, and in that case we identified the source in a known type of data artifact called “snowball” caused by large cosmic ray impacts.

In general, I am skeptical that the emission lines should show so much differences from each other in kinematics; e.g., that H α should be blueshifted by so much. The authors should at least attempt a global fit of all lines, or at least define 2 or 3 groups of lines to fit together with similar kinematics. This would provide better constraints on the weak lines, but also on strong lines with degeneracies (e.g., H α due to the strength of [NII]).

From the purely observational point of view, we believe that the fact that not all lines share the same kinematics is an extremely robust results, as can be seen by comparing, for example, [O II] with [O III]. As for the specific case of Halpha, we agree that it should have the same kinematics as [N II], and we have updated our treatment of these lines, as explained in detail in point b) at the beginning of this document. We also fix the kinematics of [S II] to that of [N II]. These lines have similar wavelength and ionization energy and so there is a strong physical reason to expect them to have similar kinematics.

I am consequently concerned about the estimate of \dot{M} from different species. That assumes a lot about the gas ionization and other uncertainties in different species. It is most reliable to use Balmer lines, which don't require such corrections (except for density).

The fact that different ionized species yield very consistent outflow rates confirms that the results are robust to the assumptions made (except for the electron density which affects all species equally). We agree that Balmer lines are more reliable, but in these data Halpha is heavily blended with [N II], and Hbeta is extremely faint (SNR~4). We thus prefer to show the full array of results in Figure 3 rather than relying on a single, low-SNR emission line.

2. Dust. The interpretation of the emission lines, particularly [SIII] and HeI, given in Figure 1 and the text is based primarily on arguments around dust obscuration. For this to be robust, it would need to be quantified using a wavelength-dependent extinction curve. Furthermore, there needs to be discussion of the gas extinction based on observed ionized emission lines. The authors currently seem to rely on extrapolation from stellar extinction in the SED model (which I assume includes the usual correction from stellar to gas color excess; Calzetti et al. 2000).

We've added an estimate of the relative extinction in the blue and red wings of different emission lines to the ionized outflow section. It is not possible to measure the dust extinction from gas emission lines because the only hydrogen lines to be detected are Hbeta and Halpha, and they suffer from low SNR and heavy blending, respectively, so that their ratio cannot be measured to the required accuracy.

The fits to [NII] and [SIII] seem very similar, so I don't think [SIII] has its shape due to differential dust attenuation. There are two [SIII] lines -- do they yield the same result?

The shape of [S III] is weakly affected by dust attenuation and is mostly determined by the outflow velocity profile projected along the line of sight (it is the shape of [O III] that

we claim to be determined by differential dust attenuation). The fainter [S III] line has a very low SNR and we cannot reliably measure its kinematics.

There are similarities between the [S III] and [N II] kinematics, even though [N II] has a lower velocity dispersion. This suggests that the observed [N II] emission may be due, at least in part, to the outflowing gas. Given the similar ionization energy of [N II] and [O II], one could ask why the [N II] profile is broader. The answer is likely a combination of SNR ([N II] is much stronger than [O II] in this galaxy) and dust attenuation: the red wing of [N II]6585 is driving the velocity dispersion to large values in the fit, which would be much harder to see in [O II] since the red wing there is heavily attenuated. However, the [N II] profile is blended with H α which makes it impossible to draw strong conclusions, and this is why we do not use the [N II] kinematics in our analysis.

The He I is interesting, and I like the interpretation -- have the authors double-checked for wavelength calibration issues?

When we first discovered the He I velocity shift we thought about a wavelength calibration issue, but we ruled out this possibility after conducting extensive tests. Moreover, the Pa gamma absorption line, which is right next to He I, appears to be exactly at the expected wavelength, confirming that the observed velocity shift of He I is real.

3. Velocities. The manuscript uses more-or-less a "maximum" velocity as the bulk velocity of the gas. But most of the gas, as detected, is at the central velocity of each line. This is what is appropriate for calculating mass outflow rates. There may be some unresolved projection effects, though not in the model as presented. Thus the outflow rates will be lower by a factor of a few.

We have updated the outflow velocity definition, as explained in point a) at the beginning of this document.

4. Size. It is not obvious from Figure 8 that the highly-blueshifted [O III] line is spatially-resolved. The size the manuscript suggests appears to use the size of the less blueshifted part of the line. This circles back to the point about line fitting above -- are there constraints on the spatial extent of the other observed lines? Are there two outflows here, one lower-velocity, spatially resolved and one high-velocity, unresolved?

In our model, the mass-conserving outflow is denser near the galaxy, and so most of the emission should come from the central regions, while the fainter emission from the outer regions should have a low line-of-sight velocity purely for projection effects. However,

we find the two-outflows scenario proposed by the referee quite convincing, and we have added it to the discussion of the ionized outflow. Some of the other emission lines are also spatially resolved (with a similar size to that measured for [O III]) but their SNR is not sufficient for a detailed analysis.

5. Presence of an AGN. The presence of an AGN is inferred from the galaxy's position on the line ratio diagrams in Figure 9. However, the only contextualization is comparison to low- z galaxies, for which this system appears as an extreme outlier. A robust comparison to recent results on both high- z star-forming galaxies and AGN is warranted. Galaxies shift on these diagrams with increasing redshift. Simply comparing to low- z data and models is insufficient. One indicative point in favor of an AGN is the combination of the strong blueshift of [OIII] with the high-velocity component being unresolved.

We have added to Figure 9 the sample of $z \sim 2$ galaxies from the MOSDEF survey and the sample of AGNs observed with JWST/NIRSpec by the GA-NIFS survey.

6. Covering fraction. The covering fraction can in fact be determined from the existing data; see, e.g., the formulae in Section 4.3 of Hamann et al. 1997, ApJ, 478, 80.

We thank the referee for the suggestion and the reference. We now state that the Ca II doublet is consistent with a covering fraction of unity; however the large error bar on the weak Ca II H line ($\sim 25\%$ uncertainty) does not allow us to put a strong constraint on the covering fraction. In practice, this does not add much to the result obtained by looking at the maximum absorption in any line, which already gives $C_f > 0.5$.

7. Absorption line fitting: The model for CaII shown in Figure 6 does not look like a 2-to-1 optical depth ratio -- it would be better to fix it to 2, or at the very least set that as an upper limit for the ratio, since it physically can't be larger than that in the optically-thin limit.

We followed the referee's suggestion and fixed the Ca K/H ratio to 2. This has minimal effects on the results.

The authors assume that NaI is optically thin based on CaII. However, NaI is typically mildly optically thick when observed in these contexts. CaII H and K are not usually observed, so the fact that the gas is dense enough to detect it suggests that NaI is even more likely

to be optically thick. (The difference in detection rate presumably arises due to the large difference in dust depletion, given that they have similar oscillator strengths.) A fit with the line ratio allowed to vary would be more appropriate. If in fact the optical depth is higher, the covering factor may differ from unity.

We agree that Na I could be optically thick, and we note in the text that our neutral gas mass should be considered a lower limit because of this. This is a conservative assumption, because in the optically thick case the true neutral gas mass would be higher than our estimate, thus strengthening the conclusions of our study regarding the ability of the neutral outflow to quench the galaxy. It is not possible to measure the Na D doublet line ratio from our data because, since the two lines are blended, the line ratio is degenerate with the velocity shift.

The fact that Ca II absorption is not usually detected does not have a straightforward interpretation, because it may also depend on the added complication of modeling the underlying stellar absorption not only in Ca H and K, but also in H ϵ . We thus prefer avoiding a detailed discussion of optical depth and covering fraction, which is outside the scope of our study, and maintain simple, conservative assumptions.

The optical depth and blueshift of Na I D may also be a lower and upper limit, respectively, if in fact there is redshifted resonant emission that is filling in the red side of the absorption line. The presence of redshifted He I could suggest that such emission is a real possibility.

We added this caveat in the text when discussing the measurement of the absorption lines.

Minor textual point:

- Please quantify the statement "relatively weak emission lines" as a indicator of quiescence. At issue is really the equivalent width of the Balmer emission lines. Similarly, the manuscript states that the "deep Balmer absorption lines clearly indicates that the galaxy is in the `post-starburst` phase." Again, the indicator is this combined with low Balmer emission line equivalent width, quantitatively.

We have removed the statement on the "post-starburst" phase for lack of space. The statement about the relatively weak emission lines is supposed to be qualitative, to guide the interpretation of the spectrum for non-experts. We prefer to avoid cluttering the introductory part of the article with quantitative measurements that would be difficult

to interpret for most readers. Instead, we devote an entire section in the Methods to the estimate of the star formation rate using different methods.

Reviewer Reports on the First Revision:

Referees' comments:

Referee #1 (Remarks to the Author):

The authors made a reasonable effort to address all the points I raised in my first report. I feel that this manuscript is now acceptable for publication.

Referee #2 (Remarks to the Author):

I thank the authors for their close attention to my first report, and that of my fellow referee.

The authors have made a good case in their reply for the significance and originality of their results. They have also made an effort to address some of my concerns about the line fits and size of the outflow. As the authors state, the details of the fits and exact adopted outflow radius will not impact this significance and originality.

That said, I continue to be uncomfortable with the line fits and the quite divergent results for different emission lines. The authors have attempted a self-consistent model for this (Figure 2). I can buy most of this model, but I find myself skeptical about component (4) of this scenario. And trying to fit the other lines into this scenario (Extended Table 1 and Extended Figure 4) is frankly, difficult. E.g., I don't understand at all why H α and [NII] should not show a blueshift as observed in some other lines (except for [OIII], which is so often more blueshifted in AGN). Or, e.g., that H α and H β should show such different velocities -- I've never seen that before, even in very dusty systems. Aside from [OIII] (and HeI, for which the authors have made a convincing atomic physics argument), it is not at all typical for the other lines shown to be so divergent in velocity and linewidth. As I mentioned in my first report, I would like to see the authors attempt a global fit to all lines, though obviously [OIII] and HeI could have a different velocity/linewidth.

Going back to outflow size -- if [OIII] is so highly blueshifted in the nucleus, and this is where most of the flux is, then the appropriate outflow size for the [OIII] outflow is not 3 kpc but whatever is indicated by the blueshifted nuclear flux. There is a less blueshifted second component (this seems to be the origin of the 3 kpc size) which really needs to be treated separately because the velocity and linewidth are quite a bit smaller, and the size is quite a bit larger.

Are there any other lines which are indicative of the size of the line-emitting gas? E.g., [NII] and H α appear quite bright, comparatively. Do they show extended emission, and, if so, is it blueshifted as the [OIII] extended gas appears to be?

This of course all connects to the absorbing gas size, as well, which the paper infers from the ionized gas. E.g., if the absorbing outflow is comparable in size to the small, high-velocity [OIII] outflow, then the size could be as small as the stellar emission, which the paper quotes as 0.6 kpc for the half-light radius. Which of course would reduce the outflow rate similarly. I'm not suggesting that is the appropriate size to use, but I think it needs to be clearer in the text that this is a possibility, at least without a direct constraint on the neutral gas outflow size.

Finally, I thank the authors for changing their neutral velocity definition. However, where the emission lines are so blueshifted (like [OIII]), $v_{\text{center}} + 2\sigma$ doesn't seem like the best choice -- v_{center} better represents the bulk of the blueshifted side of the outflow, as the authors argue in their model for the [OIII] emission. If, on the other hand, the line is tracing both sides of the outflow (as in component 4 of the model) then $v_{\text{center}} + 2\sigma$ may be appropriate.

Author Rebuttals to First Revision:

I thank the authors for their close attention to my first report, and that of my fellow referee.

The authors have made a good case in their reply for the significance and originality of their results. They have also made an effort to address some of my concerns about the line fits and size of the outflow. As the authors state, the details of the fits and exact adopted outflow radius will not impact this significance and originality.

That said, I continue to be uncomfortable with the line fits and the quite divergent results for different emission lines. The authors have attempted a self-consistent model for this (Figure 2). I can buy most of this model, but I find myself skeptical about component (4) of this scenario. And trying to fit the other lines into this scenario (Extended Table 1 and Extended Figure 4) is frankly, difficult. E.g., I don't understand at all why H α and [NII] should not show a blueshift as observed in some other lines (except for [OIII], which is so often more blueshifted in AGN). Or, e.g., that H α and H β should show such different velocities -- I've never seen that before, even in very dusty systems. Aside from [OIII] (and HeI, for which the authors have made a convincing atomic physics argument), it is not at all typical for the other lines shown to be so divergent in velocity and linewidth. As I mentioned in my first report, I would like to see the authors attempt a global fit to all lines, though obviously [OIII] and HeI could have a different velocity/linewidth.

As requested by the referee, we have performed a global fit, assuming that all lines have identical redshift and identical line width, except for [O III] and He I. The figure below compares the original fit in our submitted manuscript (model *a*) to the new global fit (model *b*). The third column shows another type of global fit (model *c*), where we fix the kinematics separately for high-ionization lines ([O III], [Ne III], [S III]) and low-ionization lines, but still leave He I free. [S II] and He I are not shown in the figure because their treatment does not vary across the three methods (since the [S II] kinematics were already fixed to those of H α in model *a*).

The most notable result of this comparison is that [O II] is definitely not well fit in the “global” models compared to the “free” model — in particular, the line width of [O II] is substantially narrower than that of [N II] and H α . This comparison is complicated by the possible presence of a broad H α emission due to the BLR, as we discuss in the manuscript; nonetheless this remains a clear indication that the assumption of identical kinematics for all low-ionization lines is too simplistic.

The [Ne III] line has low SNR, and it is more difficult to draw robust conclusions from this comparison. Model *b* appears comparable to, or slightly worse than, model *a*; however there are no physical reasons to believe that this line, which requires a ionization energy higher than that needed for [O III], should have the same kinematics as the low-ionization lines. Model *c* represents the physically motivated choice, but this model yields a rather bad fit. It is possible that the two pixels with negative flux in the blue wing of the line are actually due to bad data reduction (this might become clear with future versions of the pipeline), however for the moment we must conclude that this

a) Original fits, free kinematics

b) Same kinematics for all lines except [O III]

c) Separate kinematics for low- and high-ionization

fit is substantially worse than the others. The flux values measured by the three fits differ from each other by less than 10%, a variation that is negligible compared to the statistical uncertainties.

The **Hbeta** line also has a low SNR; in this case all three fits appear similarly good. Our original model (model *a*) yields a velocity shift and line width that are respectively 1-sigma and 2-sigma away from the Halpha kinematics. This means that we cannot make any strong claims regarding the difference in kinematics between Halpha and Hbeta. We agree that the observational evidence is too weak to claim that Hbeta is blueshifted, and that fixing the Hbeta kinematics to those of Halpha (and [N II]) is preferable. The Hbeta flux measurement is consistent across the three fitting methods to within 3%.

The lines with the highest SNR in our data are **[O III]** and the **Halpha + [N II]** complex. The three models yield nearly identical results for these lines due to their high SNR. So, while we are formally fitting multiple lines simultaneously, in practice we are fixing the kinematics of some lines to those of [O III] and the kinematics of other lines to those of Halpha + [N II].

Finally, for **[S III]** the three fits appear to be comparable, even though the blueshift predicted by model *c* is too large for the observed data. The [S III] flux measurement is consistent across the three fitting methods to within 16%.

The main results of this exercise are the following: 1) different assumptions on the line kinematics leave the line fluxes unchanged (within the statistical uncertainty); 2) we cannot make strong claims on the kinematics of the faint lines because they are too uncertain. We are now in a position to address the specific points raised by the referee:

- 1) First of all, the referee brings up several times the fact the line kinematics are highly unusual. We agree with this statement, which in our opinion is not unexpected given the unusual (or, rather, unique) nature of 11142: a massive galaxy experiencing rapid quenching, with very little star formation activity left but with a relatively large AGN-driven outflow. The vast majority of published studies in this redshift range have targeted gas-rich systems (quasars, starbursts, or main-sequence galaxies) with much stronger emission lines. The lack of a strong component due to star formation makes the spectrum of 11142 different from most of the published spectra at $z \sim 2$.
- 2) We acknowledge that we have over-interpreted the Hbeta kinematics, particularly in the first submitted version of the manuscript. We believe it is possible that Hbeta is blueshifted, but given the low SNR and the large statistical uncertainties on the measured kinematics, we consider this more of a speculation than a measurement. In the updated manuscript we now fix the Hbeta kinematics to those of Halpha and [N II], similarly to what we were doing for [S II]. We point out that the velocity dispersion for Halpha and [N II] is large, 465 km/s, consistent with outflowing material — this means that the Halpha and [N II] line fluxes could also be used, in principle, to derive a mass outflow rate. However these lines are heavily blended

(and potentially contaminated by BLR emission) and we thus avoid using them in any quantitative result.

- 3) *“I don't understand at all why H α and [N III] should not show a blueshift as observed in some other lines (except for [O III])”*. We now simply state that low-ionization lines are all at systemic velocity (H β , H α , [N II], and also [O II]). The only blueshifted lines are high-ionization lines in the blue part of the spectrum, i.e. [Ne III] and [O III]. At longer wavelengths (He I and [S III]) things are different, as discussed below. We believe this is the most natural explanation for the observed lines.
- 4) The referee is skeptical regarding our interpretation for the [S III] emission, but we argue that our model is physically motivated. [S III] happens to be very close to He I both in ionization energy (23.3 eV vs 24.6 eV) and in wavelength (0.95 μ m vs 1.08 μ m): this means that both lines originate in the same type of gas, and experience similar dust attenuation. While the blue side of He I is suppressed by resonant scattering, the red side is not affected by this phenomenon: this means that the red side of He I and the red side of [S III] must behave very similarly. We thus conclude that *[S III] should present a relatively broad red side tracing the receding outflow, just like He I*. Moreover, since [S III] is not affected by resonant scattering, it should also present a broad blue side tracing the approaching outflow (which we know exists because of the [O III] profile); the blue side can be slightly stronger than the red side because it does not experience dust attenuation. This is confirmed by our data, where [S III] has a roughly symmetric profile (actually with a slightly stronger blue side), and a line width that is larger than what expected simply from the galaxy dynamics. To conclude, we believe that the cartoon shown in Figure 2 is the simplest model that can explain self-consistently the observed line kinematics. If we attribute the He I profile to resonant scattering, then the interpretation of [S III] as tracing both the approaching and the receding side of an outflow is necessary.

In the updated manuscript, we now give a brief overview of the line kinematics according to our simple physical model. We fix the H β kinematics to that of H α and [N II] rather than performing a simultaneous fit because this yields the most straightforward interpretation (i.e., the kinematics are really measured only from H α and [N II], and then we choose to apply them to other lines as well). We maintain [Ne III] free because fixing it to the [O III] kinematics leads to a bad fit, and also because the different wavelength and ionization energy may explain small intrinsic differences in the kinematics of these two lines. Finally, we leave the [S III] kinematics free because this leads to results that are consistent with our physical model (i.e. no velocity shift and large width). Additionally, we now mention the fact that the line fluxes are very stable to assumptions on their kinematics, as revealed by these new fits. This is an important result, since the line fluxes are used to derive the mass in the ionized outflow from four different transitions, which all yield consistent results.

Going back to outflow size -- if [OIII] is so highly blueshifted in the nucleus, and this is where most of the flux is, then the appropriate outflow size for the [OIII] outflow is not 3 kpc but whatever is indicated by the blueshifted nuclear flux. There is a less blueshifted second component (this seems to be the origin of the 3 kpc size) which really needs to be treated separately because the velocity and linewidth are quite a bit smaller, and the size is quite a bit larger.

It is possible, as the referee suggests, that there are two separate components in the [OIII] outflow, but this is not the only explanation. For example, in the simplest case of a spherical outflow, we expect to find small velocity shift and line width in the outer regions of the outflow, where the velocity of the gas is mostly perpendicular to the line of sight. For the case of a conical outflow, we refer to Carniani et al. (2015), who developed a more realistic, dust-obscured outflow model to make mock IFU observations, shown in the figure below. We placed a representative slit (in black and white) on the figure to illustrate what would be seen on a 2D spectrum. The predictions of this model are qualitatively in agreement with our observations: the nuclear region has larger flux, blueshift, and line width compared to the outer regions. The Carniani model assumes that the PSF is larger than the outflow, which is not true in our case, but this has no effect on the qualitative comparison. Clearly, the observed properties of [O III] do not necessarily require the presence of a second outflow component with different size and velocity. We believe that the single-outflow explanation is simpler and therefore more likely, but we definitely cannot rule out the two-component scenario. We had already added a brief discussion of this scenario in the manuscript during the first round of revisions; we have now slightly updated that text for clarity. In the two-component scenario, we are unable to provide an estimate of the fast component because it is not spatially resolved in our data.

[Redacted text and figure]

[Redacted text and figure]

Are there any other lines which are indicative of the size of the line-emitting gas? E.g., [NII] and H α appear quite bright, comparatively. Do they show extended emission, and, if so, is it blueshifted as the [OIII] extended gas appears to be?

In addition to [O III], the only lines where we can attempt a spatially resolved study are [O II] and [N II] + H α . In order to perform this analysis it is necessary to carefully subtract the stellar emission, because these lines are not as bright as [O III], and overlap in wavelength with strong Balmer absorption lines. We build a 2D model of the stellar emission by combining the spatial profile of the 2D spectrum with the Prospector best fit of the stellar spectrum (this is a more realistic model than what done for [O III]). After subtracting this model from the data, we are able to detect spatially extended emission in [O II], [N II], and H α ; see figure below. All three lines show extended emission on the top side of the 2D spectrum, with a size of ~ 4 pixels, similar to what seen in [O III]. This definitely confirms the presence of ionized gas at a distance of ~ 3 kpc from the center, substantially larger than the stellar size. Moreover, in all lines this extended emission is slightly blueshifted. The [N II] and H α lines are blended, thus making it difficult to interpret the line morphology. The [O II] line offers a clearer view: most of the ionized emission comes from the nucleus, consistent with our analysis of the 1D spectrum, but we also detect faint blueshifted emission, both in the nucleus and in the outer regions (this emission is also visible as a small blue bump in the 1D spectrum). The morphology of this fainter emission resembles that observed in [O III], but with different brightness ratio between the nuclear and outer regions. A detailed analysis of these aspects is beyond the scope of this work, and a conclusive interpretation of the outflow geometry would require integral field spectroscopy.

We conclude that the analysis of the [O II] and [N II] lines confirms the size estimate of $R \sim 3$ kpc for the ionized outflow, and we added this statement to the manuscript.

This of course all connects to the absorbing gas size, as well, which the paper infers from the ionized gas. E.g., if the absorbing outflow is comparable in size to the small, high-velocity [OIII] outflow, then the size could be as small as the stellar emission, which the paper quotes as 0.6 kpc for the half-light radius. Which of course would reduce the outflow rate similarly. I'm not suggesting that is the appropriate size to use, but I think it needs to be clearer in the text that this is a possibility, at least without a direct constraint on the neutral gas outflow size.

We have added another argument in favor of a large size for the neutral outflow, based on the results of Rudie et al. (2017 ApJ, 843, 98), who report the detection of neutral gas at a projected distance of 38 kpc from a massive, quenching galaxy at $z \sim 2$. Nonetheless, we agree that the possibility of a much smaller neutral outflow size must be considered; we already address this in the manuscript, at the end of the “Neutral Outflow” section. As we explain in the text, this scenario requires a different definition of the outflow velocity, since in this case the observed neutral gas is not in a large, thin shell. We obtain a minimum mass outflow rate of 11 M_{sun}/yr , which would still be larger than the ionized rate and the residual star formation rate, thus confirming our main results (this minimum value also includes a conservative choice for the amount of neutral gas not associated with the outflow, as explained in the text).

Finally, I thank the authors for changing their neutral velocity definition. However, where the emission lines are so blueshifted (like [OIII]), $v_{\text{center}} + 2\sigma$ doesn't seem like the best choice — v_{center} better represents the bulk of the blueshifted side of the outflow, as the authors argue in their model for the [OIII] emission. If, on the other hand, the line is tracing both sides of the outflow (as in component 4 of the model) then $v_{\text{center}} + 2\sigma$ may be appropriate.

The referee's suggestion ($v_{\text{out}} = v_{\text{center}}$) is correct only if the conical outflow has a narrow opening angle and is directly pointing toward the observer. For the general case shown in our Figure 2, the observed v_{center} is clearly smaller than the true outflow velocity due to an inclination correction. The relation between the true outflow velocity and the observed line kinematics is complex and depends on unknown parameters such as inclination, opening angle, and dust attenuation (see e.g. Harrison et al. 2012 MNRAS, 426, 1073 and Liu et al. 2013 MNRAS, 436, 2576). Our choice of taking the “most extreme” observed velocity, i.e. $v_{\text{out}} = v_{\text{center}} + 2\sigma$, is admittedly too simplistic. We now take an intermediate value: $v_{\text{out}} = v_{\text{center}} + 1\sigma$, and consider a systematic uncertainty on the outflow velocity equal to 1 sigma, so that both scenarios ($v_{\text{out}} = v_{\text{center}}$ and $v_{\text{out}} = v_{\text{center}} + 2\sigma$) are included in the uncertainty. We apply this definition to the blueshifted lines, i.e. [O III] and [Ne III]; while we maintain the 2σ definition for “centered” lines (i.e. with $v_{\text{center}} \sim 0$), which now include Hbeta in addition to [S III]. As we mention in the revised text, the exact definition of the outflow velocity has a very small effect on the final estimate of the mass outflow rate, considering the large systematic uncertainties involved in the calculations.

Finally, we have updated the abstract, following editorial suggestions, and added small updates and corrections to the text.

Reviewer Reports on the Second Revision:

Referees' comments:

Referee #2 (Remarks to the Author):

I thank the authors for their attention to the details of my second report. I can now fully recommend the paper for publication.

Author Rebuttals to Second Revision:

March 28th, 2024

This is the final revision for the manuscript. The last referee report did not request any further change, so this version is virtually identical to the last submission.

I wish to participate in transparent peer review.

Sirio Belli